# Progressive Data Dropout: An Embarrassingly Simple Approach to Train Faster

**Shriram M S[1]\*, Xinyue Hao[2]\*, Shihao Hou[3], Yang Lu[3], Laura Sevilla-Lara[2],**
**Anurag Arnab, Shreyank N Gowda[4]**
[1]Department of Computer Science, University of Manchester
[2]School of Informatics, University of Edinburgh
[3]School of Informatics, Xiamen University
[4]School of Computer Science, University of Nottingham

## Abstract

The success of the machine learning field has reliably depended on training on large datasets. While effective, this trend comes at an extraordinary cost. This is due to two deeply intertwined factors: the size of models and the size of datasets. While promising research efforts focus on reducing the size of models, the other half of the equation remains fairly mysterious. Indeed, it is surprising that the standard approach to training remains to iterate over and over, uniformly sampling the training dataset. In this paper we explore a series of alternative training paradigms that leverage insights from hard-data-mining and dropout, simple enough to implement and use that can become the new training standard. The proposed Progressive Data Dropout reduces the number of effective epochs to as little as 12.4% of the baseline. This savings actually do not come at any cost for accuracy. Surprisingly, the proposed method improves accuracy by up to 4.82%. Our approach requires no changes to model architecture or optimizer, and can be applied across standard training pipelines, thus posing an excellent opportunity for wide adoption. Code can be found here: `https://github.com/bazyagami/LearningWithRevision`.

## 1 Introduction

Advances in deep learning over the last years have consistently come at the cost of substantial computational resources. Each year, the computational cost of the most notable models multiplies by over 4 times Epoch AI [2023]. This poses challenges both in terms of real-world applications and sustainability Gowda et al. [2023].

In response to these challenges, a significant body of research has emerged focused on developing efficient neural architectures. Models such as MobileNet Howard et al. [2017], Sandler et al. [2018], EfficientNet Tan and Le [2019], and EfficientFormer Li et al. [2022] have demonstrated that carefully designed architectures can achieve competitive performance with significantly reduced parameter counts and computational demands. These architectural innovations have been complemented by techniques such as pruning Han et al. [2015], quantization Jacob et al. [2018], and knowledge distillation Hinton et al. [2015], all aimed at reducing the resource footprint of deep neural networks.

While most efficiency efforts focus on the model, another underexplored direction is improving *how* we train. Standard training treats all samples equally across all epochs, but not all data points are equally useful throughout the learning process. Some examples are consistently misclassified and carry high learning value, while others are confidently and correctly classified early on and may

---

\* Equal Contribution.

39th Conference on Neural Information Processing Systems (NeurIPS 2025).

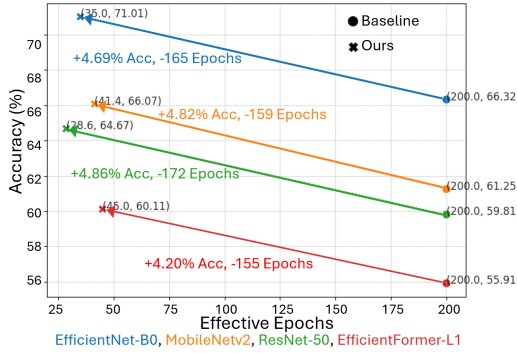

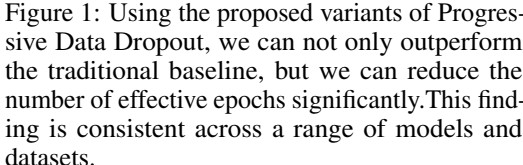

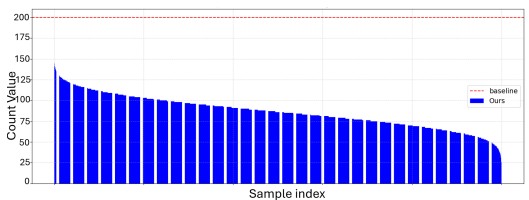

Figure 1: Using the proposed variants of Progressive Data Dropout, we can not only outperform the traditional baseline, but we can reduce the number of effective epochs significantly. This finding is consistent across a range of models and datasets.

Figure 2: A possible reasoning for the effectiveness of our approach is that random dropout maintains representative coverage and introduces beneficial stochasticity. Here, we see an example of the number of times each sample goes through back-propagation. Traditionally, each sample would go through this 'total epochs' number of times (red line).

contribute little to further model improvement Katharopoulos and Fleuret [2018]. Allocating equal computation to both types may be wasteful. Human learning strategies offer valuable insights for improving machine learning systems. When studying for examinations, humans rarely allocate equal attention to all material. Instead, they often adopt strategies that prioritize challenging concepts while ensuring comprehensive knowledge through periodic revision Dunlosky et al. [2013], Bjork et al. [2011]. This selective focus on difficult content, followed by holistic review, allows for efficient allocation of cognitive resources while maintaining broad coverage of the subject matter. A parallel in deep learning is dropout Srivastava et al. [2014], where neurons are randomly deactivated during training to prevent co-adaptation and improve generalization. However, dropout operates on the model's activations rather than the data. In contrast, data dropout Wang et al. [2018], the idea of selectively dropping training samples during learning, remains relatively unexplored.

In this paper, we introduce Progressive Data Dropout (PDD), a simple yet surprisingly effective family of strategies that progressively drops subsets of training data across epochs. Our methods are extremely easy to implement and grounded in both cognitive principles and practical efficiency goals. We explore three variants: a difficulty-based approach that initially trains only on hard examples; a scalar-based approach that randomly drops a fixed proportion of data each epoch; and a hybrid strategy that drops random samples using the same schedule as the difficulty-based method.

Beyond the cognitive inspiration, our approach offers significant practical benefits. By initially focusing on difficult examples, the model processes substantially fewer data points during early epochs, leading to reduced computational requirements. This selective focus not only accelerates training but also appears to guide the model toward more generalizable representations, as evidenced by improved performance at test time. Our experimental results demonstrate that models trained with our approach achieve comparable or superior accuracy while requiring only **0.124×** the number of effective epochs compared to standard training procedures. This efficiency gain is particularly notable for already-efficient architectures like MobileNet and EfficientFormer, where further reducing training costs can significantly impact deployment scenarios with constrained resources. We plot some of these results on CIFAR-100 Krizhevsky and Hinton [2009] in Figure 1.

Our work makes the following key contributions:

- We propose *Progressive Data Dropout*, a simple and general method to reduce training cost by progressively dropping subsets of data across epochs, inspired by cognitive learning strategies and model regularization techniques.

- We design and investigate three dropout variants: (1) a difficulty-based schedule that retains only hard examples early in training, (2) a scalar-based random sampling method, and (3) a schedule-matched random dropout strategy that surprisingly yields the best accuracy.

- Our approach significantly reduces effective epochs by up to 89.5% fewer epochs compared to standard training, whilst still improving or matching accuracy, demonstrating effectiveness

on multiple architectures including EfficientNet, MobileNet-V2, ResNet, EfficientFormer and ViT-B-MAE. We also show the potential cost saving in self-supervised learning.

- We show that Progressive Data Dropout is compatible with standard supervised training pipelines and can be easily integrated without architectural or optimization changes.

## 2 Related Work

Several learning paradigms share a similar underlying goal with our proposed method, namely, improving training efficiency and generalization, though they differ significantly in implementation and motivation. Curriculum learning, active learning, and data pruning are all relevant lines of research that intersect with our objectives. While we briefly discuss these in the below to contextualize our contributions, we focus our detailed comparisons on methods that explicitly implement forms of data dropout, given the page constraints.

**Model Compression and Training Efficiency.** Significant progress has been made in making neural networks more efficient at inference time through model compression techniques. Pruning Han et al. [2015], quantization Jacob et al. [2018], and knowledge distillation Hinton et al. [2015] reduce the size or precision of model weights, allowing faster inference with minimal accuracy loss. Parallel efforts in efficient architecture design, such as MobileNet Howard et al. [2017], EfficientNet Tan and Le [2019], and EfficientFormer Li et al. [2022], focus on reducing the number of parameters and FLOPs during inference. In contrast, our work targets the training phase rather than inference, aiming to reduce effective epochs and data usage dynamically, independent of the underlying model architecture.

**Curriculum Learning and Sample Importance.** Curriculum learning Bengio et al. [2009] and hard example mining Shrivastava et al. [2016] propose organizing training data to emphasize samples based on their difficulty or relevance. These methods aim to present samples in a structured order to accelerate convergence or improve performance. More recent work Katharopoulos and Fleuret [2018] studies the informativeness of data during training and suggests adaptive data selection to optimize learning. Our work differs in that we progressively drop samples rather than reordering or reweighting them. Furthermore, we introduce both difficulty-based and random dropping strategies, showing that even unstructured data dropout can lead to improved generalization.

**Training on Subsets and Data Pruning.** Several recent works investigate training on a reduced subset of the data to minimize computational overhead. Data Diet Toneva et al. proposes filtering data based on metrics like gradient norm and error L2-norm to identify examples that are likely to be forgotten during training. Dataset pruning methods, such as those proposed by Yang et al. Yang et al. [2022], aim to identify a minimal subset of training data that closely preserves the generalization performance of the full dataset, using optimization techniques with theoretical guarantees. Coleman et al. Coleman et al. introduce a "selection via proxy" (SVP) approach that accelerates data selection by training small proxy models instead of the full-scale target network, enabling faster core-set selection or active learning. While related in goal, our method differs in important ways. Many of these approaches require substantial pre-processing or side-training before model training begins, for example, Data Diet requires tracking forgetting events early in training Toneva et al., and SVP uses proxy models trained in advance Coleman et al.. Influence-based selection methods such as Koh and Liang [2017], Feldman and Zhang [2020] estimate the effect of removing samples on model predictions, which can be computationally expensive. Score-based heuristics (e.g., confidence scores or gradient magnitudes) also require repeated forward or backward passes over the entire training set Mirzasoleiman et al. [2020], Mindermann et al. [2022]. In contrast, our approach is integrated directly into the training loop, progressively dropping examples over epochs without requiring separate scoring, proxy models, or influence estimation.

**Active Learning.** Active learning Settles [2009] aims to reduce labeling and training costs by selectively querying the most informative data points from an unlabeled pool. Common strategies include uncertainty sampling Nguyen et al. [2022], where examples with the lowest prediction confidence are chosen; margin sampling Balcan et al. [2007], which selects samples near the decision boundary; and query-by-committee Kee et al. [2018], which leverages disagreement among multiple models to identify uncertain data. These methods are typically iterative and assume access to large

unlabeled datasets alongside a labeling budget. In contrast, our approach operates in the fully supervised regime and does not require querying new data, rather, we progressively drop portions of the labeled dataset during training to reduce computation and potentially improve generalization.

**Data Dropout and Data-Centric Regularization.** Dropout Srivastava et al. [2014] is a widely-used regularization method that randomly drops activations during training to prevent co-adaptation and overfitting. This concept has inspired architectural variants such as DropConnect Wan et al. [2013] and stochastic depth Huang et al. [2016]. While these methods manipulate the network topology, relatively fewer studies have explored analogs at the data level. Most closely related to our method are approaches that drop data samples during training in order to improve efficiency or model generalization. These *data dropout* techniques differ in motivation and implementation, ranging from data augmentation to importance-based filtering. Some methods focus on data-level perturbations for augmentation purposes. For instance, Generalized Dropout Rahmani and Atia [2018] applies dropout directly to image pixels, generating additional variations of the same training sample to improve generalization. While effective as a form of regularization, these approaches do not aim to reduce the number of training iterations or epochs. Other works emphasize importance-based data pruning. Katharopoulos and Fleuret Katharopoulos and Fleuret [2018] propose a sampling strategy that prioritizes high-gradient or high-loss examples, dropping low-impact samples to accelerate convergence. Wang et al. Wang et al. [2018] introduce a two-round training scheme in which training samples are evaluated after an initial round and permanently dropped if they are deemed to negatively influence generalization. Similarly, DropSample Yang et al. [2016] and Greedy DropSample Yang et al. [2020] propose sample selection mechanisms that adjust data retention dynamically, particularly in the context of Chinese character classification. Zhong et al. Zhong et al. [2021] propose Dynamic Training Data Dropout (DTDD), which filters noisy samples over several epochs to improve robustness in deep face recognition. In contrast to all these methods, our approach is integrated directly into the training loop and operates in a progressive manner. Rather than relying on pretraining, expensive influence function estimation (which require second-order gradients), or permanently dropping samples after a single evaluation, we iteratively modify the training set across epochs using cognitively inspired and stochastic strategies. This allows us to balance efficiency and accuracy without the need for auxiliary models, validation sets, or expensive influence computation.

# 3 Method

We introduce *Progressive Data Dropout*, an embarrassingly simple yet effective method to accelerate neural network training by progressively reducing the training set across epochs. Inspired by cognitive science and dropout regularization, our approach incrementally drops training samples based on either difficulty or randomized criteria, while ensuring full-data coverage in the final epoch. In this section, we describe three variants of our method. A full algorithm explaining the approach can be found in the supplementary material.

## 3.1 Overview

Let $\mathcal{D}_0 = \{(x_i, y_i)\}_{i=1}^N$ be the original training dataset. At each epoch $t$, we construct a subset $\mathcal{D}_t \subseteq \mathcal{D}_0$ such that fewer samples are used as training progresses. This reduction can be based either on model confidence (difficulty-based) or through random sampling. In all variants, the final epoch uses the full dataset $\mathcal{D}_0$, mimicking the "revision" phase in human learning where comprehensive review reinforces generalization.

## 3.2 Variant 1: Difficulty-First Training

This cognitively motivated strategy Bjork et al. [2011] focuses on "hard" examples early on and uses all examples at the last epoch as a form of 'revision'. After each epoch, we evaluate the model's confidence or correctness on each sample:

- Let $p_t(x_i)$ be the model's predicted probability for the correct class of sample $x_i$ at epoch $t$.
- A sample is considered "easy" if $p_t(x_i) \geq \tau_t$, where $\tau \in [0, 1]$ is a fixed threshold .
- The dataset for the each epoch is:
$$\mathcal{D}_t = \{(x_i, y_i) \in \mathcal{D} \mid p_t(x_i) < \tau\} \ \forall t = 0, .., T$$

- In the final epoch, we reset to $\mathcal{D}_T = \mathcal{D}_0$.

This approach mimics human study habits: initially focusing on difficult material, then performing a complete review. We refer to this method as **Difficulty-Based Progressive Dropout** (DBPD).

### 3.3 Variant 2: Scalar-Based Random Dropout

This variant eschews difficulty estimation and instead uses a scalar decay factor $\alpha \in (0,1)$ to progressively reduce the dataset:

- The number of samples in epoch $t$ is $|\mathcal{D}_t| = \alpha^t \cdot N$.
- $\mathcal{D}_t$ is created by randomly sampling (without replacement) from $\mathcal{D}_0$.
- The full dataset is used in the final epoch: $\mathcal{D}_T = \mathcal{D}_0$.

This method, which we call **Scalar Random Dropout** (SRD), is the simplest to implement and requires no per-sample evaluation. Despite its simplicity, it yields surprisingly competitive performance with substantial computational savings.

### 3.4 Is Difficulty Scoring Necessary? Variant 3: Schedule-Matched Random Dropout

To test whether explicit difficulty-based scoring is essential for effective data dropout, we design a variant that uses the same per-epoch data reduction schedule as Variant 1 (difficulty-based), but drops samples *at random* rather than based on confidence.

- Let $|\mathcal{D}_t^{(1)}|$ be the dataset size at epoch $t$ under the difficulty-based method.
- In this variant, we randomly subsample $|\mathcal{D}_t^{(1)}|$ points from $\mathcal{D}_0$ at each epoch $t$, without regard to difficulty.
- As in the other methods, the final epoch uses the full dataset: $\mathcal{D}_T = \mathcal{D}_0$.

We refer to this as **Schedule-Matched Random Dropout** (SMRD). While this approach is less practical to implement in real-time (as it relies on the difficulty-based schedule computed in a separate run), it serves as a valuable ablation to probe the importance of difficulty estimation. Surprisingly, SMRD achieves the highest test accuracy across our experiments, indicating that structured data reduction alone, without any notion of sample difficulty, can yield strong generalization benefits.

To approximate the schedule without explicitly scoring sample difficulty, we also explore a mathematical approximation of the number of samples dropped per epoch. Results of this approximation are included in the supplementary material.

### 3.5 Effective Epochs

To provide a hardware-independent measure of training efficiency, we introduce the notion of *Effective Epochs* (EE). While traditional epoch counts assume that every training sample undergoes both forward and backward passes, our method performs a forward pass on all samples but selectively applies backpropagation only to a subset. This design reduces computational cost while retaining compatibility with standard data pipelines.

We define the number of *Effective Epochs* as:

$$\text{Effective Epochs} = \frac{\text{Number of Samples through Backpropagation over entire run}}{\text{Total Number of Training Samples}}$$

This quantity reflects how many full passes over the dataset (with backpropagation) were effectively executed. It captures the true training burden independent of batch size, hardware, or epoch length. Throughout the paper, we report effective epochs as our standard efficiency metric to quantify the computational savings achieved by our dropout strategies.

# 4 Discussion

Progressive Data Dropout (PDD) improves generalization by gradually reducing the training dataset during learning. At each epoch, a portion of data is dropped, randomly or based on difficulty, forcing the model to focus on more informative examples and avoid overfitting. This acts as a form of data-level regularization, analogous to dropout on model activations Srivastava et al. [2014]. Surprisingly, random dropout outperforms difficulty-based dropout, suggesting that the benefit comes more from data reduction than selective removal. However, just randomly dropping a percentage of data does not work best (see SRD). However, having a scheduled based random dropout works best. Whilst, we try to mathematically approximate this and test, we do not obtain our best results, leaving this to future work. Difficulty-based methods may skew the training distribution or overfit to noisy samples Yang et al. [2016, 2020], while random dropout maintains representative coverage and introduces beneficial stochasticity. PDD builds on insights from dataset pruning and curriculum learning. Prior work shows that large portions of training data can be discarded with minimal or even positive effects on accuracy Toneva et al., Yang et al. [2022]. Unlike methods requiring retraining, proxies, or selection phases Wang et al. [2018], Coleman et al., PDD dynamically reduces data within a single run. Unlike curriculum learning that adds or reweights samples Bengio et al. [2009], Shrivastava et al. [2016], PDD follows an anti-curriculum, reducing data as training progresses. Its success, especially in the random variant, highlights that training with less but diverse data can improve robustness and generalization.

# 5 Experimental Analysis

## 5.1 Datasets

We conduct all classification experiments using three standard image classification benchmarks: CIFAR-10 Krizhevsky and Hinton [2009], CIFAR-100 Krizhevsky and Hinton [2009], and ImageNet Deng et al. [2009]. The CIFAR-10 and CIFAR-100 datasets each consist of 60,000 $32 \times 32$ color images, split into 50,000 training and 10,000 test images. CIFAR-10 contains 10 object classes, while CIFAR-100 spans a more fine-grained taxonomy of 100 classes. Both datasets are widely used to benchmark small and mid-sized neural architectures. We also use the ImageNet (ILSVRC2012) dataset, which contains over $1.28$ million training images and $50,000$ validation images across $1,000$ object categories. The images are of higher resolution and greater diversity compared to CIFAR datasets, making ImageNet a robust benchmark for evaluating both model capacity and training efficiency. In addition to supervised classification, we also evaluate our method in a self-supervised learning setup, using ImageNet as the benchmark dataset. This allows us to assess the generality of Progressive Data Dropout beyond supervised learning scenarios.

## 5.2 Implementation Details

All variants are implemented with minimal overhead on top of standard training loops. To avoid introducing instability, batch normalization statistics and learning rates are kept consistent with standard training procedures. For variants with data reduction, we ensure that the data loader is reset at each epoch to reflect the newly selected subset. Importantly, we do **not** alter model architectures, loss functions, or optimizers. Our method is architecture-agnostic and seamlessly integrates with any supervised training pipeline. We use popular models such as MobileNetv2 Sandler et al. [2018], EfficientNet Tan and Le [2019], Efficientformer Li et al. [2022], ResNet He et al. [2016] and ViT-B Dosovitskiy et al. [2020] for the image classification tasks. We also consider MAE He et al. [2022] and perform masked pre-training using the Scalar Dropout idea and fine-tune the model using standard fine-tuning. We follow the official implementation details for ImageNet experiments, so for example EfficientNet is run for 350 epochs, while ResNet-50 is run for 100 epochs. However, for all CIFAR-100 experiments we run all models for 200 epochs and for all CIFAR-10 experiments we run them for 30 epochs. For our CIFAR experiments, we use an AdamW optimizer with a StepLR scheduler with a step size of 1. Our initial learning rate is $0.0003$. For the self-supervised learning experiments with MAE He et al. [2022], we follow standard practice He et al. [2016] and run the pre-training phase for 800 epochs. We use a fixed batch size of 32 for all our experiments. We use 8 A100 GPUs with 40GB memory. For ImageNet fine-tuning we use 1 A100 GPU with 40GB memory. For CIFAR10 and CIFAR100, we use a 4060RTX GPU with 8GB memory.

Table 1: Accuracy (%) and effective epochs reported as *accuracy/effective epochs*. All methods besides ViT-MAE are trained from scratch unless otherwise mentioned. ViT-MAE uses the publicly available checkpoint He et al. [2022]. All results have been conducted following official implementations on timm Wightman [2019]. Best results in blue and second best in green. The last column shows the gains in accuracy and drop in effective epochs of the SMRD variant.

| | Baseline | DBPD | SRD | SMRD | Acc. gain (%) | EE saved |
|---|---|---|---|---|---|---|
| *CIFAR10* | | | | | | |
| EfficientNet-B0 | 90.41 / 200 | 88.22 / 18 | 92.07 / 84 | 90.55 / 20 | 0.14 | 10 × |
| MobileNet-V2 | 88.46 / 200 | 89.55 / 22.3 | 90.73 / 84 | 90.91 / 24.0 | 2.45 | 8.33 × |
| ResNet-50 | 87.94 / 200 | 88.42 / 23 | 89.81 / 84 | 89.12 / 22.1 | 1.18 | 9.05 × |
| EfficientFormer-L1 | 82.38 / 200 | 84.47 / 28.2 | 84.38 / 84 | 85.20 / 31 | 2.82 | 6.45 × |
| *CIFAR100* | | | | | | |
| EfficientNet-B0 | 66.32 / 200 | 67.15 / 24.8 | 66.75 / 24.8 | 71.01 / 35 | 4.69 | 5.71 × |
| MobileNet-V2 | 61.25 / 200 | 62.85 / 29.6 | 62.80 / 24.8 | 66.07 / 41.4 | 4.85 | 4.83 × |
| ResNet-50 | 59.81 / 200 | 60.13 / 16.26 | 60.83 / 24.8 | 64.67 / 28.6 | 4.86 | 6.99 × |
| EfficientFormer-L1 | 55.91 / 200 | 57.62 / 30 | 56.95 / 24.8 | 60.11 / 45 | 4.20 | 4.44 × |
| *CIFAR100 finetuned from ImageNet* | | | | | | |
| EfficientNet-B0 | 83.31 / 200 | 83.95 / 25.4 | 83.58 / 24.8 | 84.45 / 33.8 | 1.14 | 5.92 × |
| MobileNet-V2 | 74.10 / 200 | 74.35 / 22.4 | 74.22 / 24.8 | 74.73 / 36.2 | 0.63 | 5.52 × |
| ResNet-50 | 78.59 / 200 | 79.92 / 21 | 79.63 / 24.8 | 81.54 / 33.2 | 2.95 | 6.02 × |
| EfficientFormer-L1 | 85.30 / 200 | 86.74 / 36.6 | 84.78 / 24.8 | 87.51 / 51.2 | 2.21 | 3.91 × |
| ViT-B-MAE | 86.81 / 200 | 85.92 / 52.2 | 82.90 / 24.8 | 88.10 / 67.2 | 1.29 | 2.98 × |
| *ImageNet* | | | | | | |
| EfficientNet-B0 | 77.10 / 350 | 77.45 / 111.2 | 64.71 / 20 | 79.51 / 142.4 | 2.41 | 2.46 × |
| MobileNet-V2 | 71.60 / 250 | 68.42 / 54.5 | 63.85 / 20 | 73.25 / 98.0 | 1.75 | 2.55 × |
| ResNet-50 | 75.04 / 100 | 76.21 / 33.0 | 71.54 / 20 | 79.18 / 46.4 | 4.14 | 2.15 × |
| EfficientFormer-L1 | 79.11 / 300 | 79.24 / 91.5 | 69.67 / 20 | 80.25 / 106.1 | 1.14 | 2.83 × |
| ViT-B-MAE | 83.10 / 100 | 83.31 / 35.4 | 79.51 / 20 | 84.13 / 41.7 | 1.03 | 2.40 × |

## 5.3 Supervised Image Classification

In Table 1, we compare the performance of EfficientNet-B0 Tan and Le [2019], MobileNet-V2 Sandler et al. [2018], ResNet-50 He et al. [2016], EfficientFormer Li et al. [2022], and ViT-B-MAE He et al. [2022] across CIFAR-10, CIFAR-100, and ImageNet. We report results for each model under the baseline training regime as well as the three main variants of our proposed method: DBPD 0.3, SMRD 0.3, and SRD 0.95. For completeness, results for DBPD 0.7 and SMRD 0.7 are provided in the supplementary material. Among all configurations, SMRD 0.3 consistently achieves the highest accuracy across most settings, while the 0.3 threshold variants also offer the greatest reduction in effective epochs. Across datasets and architectures, our approach yields improvements in accuracy of up to **4.82**% and reduces the effective epochs by as much as **12.4**%.

## 5.4 Self-Supervised Training

Self-supervised learning is among the most computationally expensive training paradigms, often accounting for over 95% of the total training cost when including fine-tuning across multiple downstream tasks. Masked Autoencoders (MAE) He et al. [2022] have shown strong promise in learning generalizable representations and improving performance on a wide range of tasks. In this work, we focus on the ViT-MAE-B model. While the original implementation pre-trains this model for 1600 epochs, we adopt the 800-epoch variant due to memory and compute limitations. Notably, the drop in accuracy for the 800-epoch model on ImageNet is less than 0.3%, making it a reasonable compromise.

To apply Progressive Data Dropout in this setting, we use the SRD 0.98 variant, which is the simplest to adapt to a self-supervised objective. Additional results using other variants are included in the supplementary material. Following pretraining, we fine-tune the model using the official MAE procedure on ImageNet and CIFAR-100, and also evaluate it on the COCO object detection benchmark to test cross-task generalizability.

Table 2: Comparison of effective epochs and accuracy for ViT-MAE-B under standard MAE (baseline) and SRD 0.98 configurations across various tasks. Accuracy is reported as Top-1 (%) for classification tasks and mAP (%) for object detection.

| Model | Pretraining EE | ImageNet Accuracy | CIFAR-100 Accuracy | COCO $AP^{box}$ |
|---|---|---|---|---|
| Baseline | 800 | 83.10% | 86.81% | 49.9 |
| SRD 0.98 | **49.95** | 77.89% | 83.81% | 47.2 |

Table 3: Baseline trained to match the effective number of epochs (EE) used by dropout variants.

| Variant | EE | Baseline (%) | Accuracy (%) |
|---|---|---|---|
| DBPD 0.3 | 25 | 63.32 | 67.15 |
| DBPD 0.7 | 33 | 63.57 | 68.80 |
| SRD 0.95 | 25 | 63.32 | 66.75 |
| SMRD 0.3 | 35 | 63.52 | 71.01 |
| SMRD 0.7 | 46 | 63.88 | 70.75 |

Despite its simplicity, SRD reduces the total number of effective epochs from 800 to just 50, a **16×
reduction** in training steps. This reduction is especially impactful given the extreme time, memory, and energy requirements of MAE pretraining. Moreover, across all downstream tasks, our method remains highly competitive with the baseline, with only negligible loss in performance. We report these results in Table 2.

## 5.5 Comparison to State-of-the-Art

We conducted comparisons against a suite of existing methods, including DataDiet (ELN and Forget) Paul et al. [2021], IES Yuan et al. [2025], Early Stopping, and InfoBatch Qin et al.. We followed the official implementations for all baselines and used EE to evaluate efficiency. Our method outperforms all compared approaches in both accuracy and EE, with higher values indicating better performance for both metrics. The results, reported as Accuracy / EE, are obtained by fine-tuning models on CIFAR-100 (pretrained on ImageNet) for 200 epochs.

Table 4: Comparison with strong active learning baselines on CIFAR-100.

| Model | Baseline | DataDiet ELN | DataDiet Forget | IES | Early Stopping | InfoBatch | SMRD (ours) |
|---|---|---|---|---|---|---|---|
| ResNet-50 | 78.5 / 200 | 78.6 / 141 | 79.1 / 142 | 79.5 / 157 | 74.3 / 121 | 80.4 / 145 | **81.5 / 33** |
| EfficientNet | 83.3 / 200 | 82.7 / 139 | 83.0 / 139 | 83.7 / 153 | 76.6 / 132 | 84.0 / 141 | **84.4 / 34** |
| EfficientFormer-L1 | 85.3 / 200 | 84.8 / 142 | 85.0 / 140 | 85.8 / 155 | 78.2 / 125 | 86.6 / 143 | **87.5 / 51** |

## 5.6 Generalization in Language Tasks

We used GPT-2 pre-training with SRD on the FineWeb-Edu dataset Penedo et al. [2024]. The results in the Table 5 show that SRD achieved substantial computational savings, reducing both the number of training steps and wall-clock time by more than half.

Table 5: NLP task performance of SRD for small GPT-2 pretraining experiments.

| Model | PPL_val | PPL_train | Num_steps | Wall clock |
|---|---|---|---|---|
| GPT-2 (Baseline) | 29.37 | 28.50 | $2,560,000$ | 13.34 hrs |
| SRD (Threshold = 0.99) | 32.81 | 31.78 | $\mathbf{1,066,720}$ | **5.91** hrs |

## 5.7 Ablation Study

To better understand the behavior of Progressive Data Dropout, we conduct an ablation study along three dimensions: dropout threshold, training duration, and matched-step baseline comparisons. For all our experiments, we use EfficientNet-B0 trained from scratch on CIFAR-100.

**Are our efficiency gains just due to fewer effective epochs?** To ensure our improvements are not simply due to reduced training time, we train a baseline EfficientNet-B0 for the same number of epochs as the equivalent effective epoch for each dropout variant. This lets us isolate whether performance improvements come from the dropout schedule itself rather than step count alone. We report these in Table 3.

**How sensitive is difficulty-based dropout to the choice of threshold?** We vary the confidence threshold $\tau$ used for dropping samples in the difficulty-based variant (DBPD), ranging from 0.1 to 0.9

in increments of 0.1 and report these results in Table 6 , Table 7, and Table 8. Higher thresholds lead to more conservative early-stage dropout, while lower values remove more data per epoch. Hence, lower thresholds tend to have a much lower step count. To ensure a balance between accuracy and epochs, we use thresholds of 0.3 and 0.7 in all experiments. Experiments in Table 6 are conducted using EfficientNet-B0 on CIFAR-100 for 200 epochs with the final epoch being full-dataset revision. For SRD, we applied it in the most straightforward form of data dropping. Table 9 presents the results on CIFAR-100 across different thresholds for reference.

Table 6: Varying threshold on EfficientNet-B0 (CIFAR-100, 200 epochs). Threshold 0.0 = baseline.

| Threshold | 0.0 | 0.1 | 0.2 | 0.3 | 0.4 | 0.5 | 0.6 | 0.7 | 0.8 | 0.9 |
|---|---|---|---|---|---|---|---|---|---|---|
| Accuracy (%) | 66.32 | 66.61 | 66.92 | 67.15 | 68.07 | 67.53 | 68.80 | 68.89 | **69.03** | 68.93 |

Table 7: Analysis for different dropout thresholds on CIFAR-100 dataset with training from scratch.

| Threshold | 0.0 | 0.1 | 0.2 | 0.3 | 0.4 | 0.5 | 0.6 | 0.7 | 0.8 | 0.9 |
|---|---|---|---|---|---|---|---|---|---|---|
| EfficientFormer | 55.91 | 56.37 | 56.88 | 57.52 | 57.85 | 57.98 | 58.12 | 58.08 | 58.17 | 58.22 |
| MobileNet-v2 | 61.25 | 61.49 | 62.17 | 62.85 | 62.98 | 63.21 | 63.27 | 63.31 | 63.86 | 63.79 |

Table 8: Analysis for different dropout thresholds on ImageNet with training from scratch.

| Threshold | 0.0 | 0.1 | 0.2 | 0.3 | 0.4 | 0.5 | 0.6 | 0.7 | 0.8 | 0.9 |
|---|---|---|---|---|---|---|---|---|---|---|
| ResNet-50 | 75.04 | 75.55 | 75.80 | 76.21 | 76.34 | 76.42 | 76.79 | 77.14 | 77.20 | 77.19 |

Table 9: SRD Threshold sensitivity on CIFAR-100 dataset.

| Threshold | 0.0 | 0.2 | 0.4 | 0.6 | 0.8 | 0.9 | 0.92 | 0.94 | 0.95 | 0.96 |
|---|---|---|---|---|---|---|---|---|---|---|
| EfficientFormer | 55.91 | 17.84 | 27.92 | 38.85 | 48.83 | 56.15 | 56.74 | 56.91 | 56.95 | 56.97 |
| MobileNet-v2 | 61.25 | 24.41 | 35.22 | 48.63 | 57.79 | 62.18 | 62.59 | 62.77 | 62.80 | 62.82 |

**Can we shorten training and still retain performance?** To assess training-time sensitivity, we repeat all dropout variants under a 50-epoch training schedule (including final revision). We compare them to a 50-epoch baseline. Despite shorter training, all variants outperform the baseline (and the baseline trained to 200 epochs as well). Further, with just 50 epochs we are only 1.75% worse off than our variant with 200 epochs. However, the baseline drops by 2.67%. This shows that the proposed data dropout can improve generalization even under constrained budgets. The results can be seen in Table 10.

**What happens when we use more epochs for revision?** We observe that test performance often peaks shortly after the first revision epoch, followed by a gradual decline. As shown in Figure 3, the model experiences a significant boost in accuracy during the first revision but this is subsequently followed by a steady drop, eventually converging toward the performance of the baseline model. We hypothesize that this pattern arises because the model, after the initial generalization gains, begins to overfit to the full dataset, thereby losing some of its earlier generalization capability.

**How many samples are we dropping over time?** Figure 4 illustrates the number of training samples retained at each epoch under the DBPD (Difficulty-Based Progressive Dropout) strategy for two different threshold settings: 0.3 (darker line) and 0.7 (lighter line). In both cases, training begins with the full dataset. As the model improves and classifies more samples with high confidence, the number of retained examples drops sharply, particularly in the early epochs. The higher threshold (0.7) causes a more aggressive pruning of data, dropping a larger fraction of confident samples per epoch. As a result, it reaches a low sample count much earlier, within the first 15–20 epochs, leaving fewer than 10,000 samples for much of training. In contrast, the 0.3 threshold maintains more samples for a longer duration, leading to a more gradual decay. Both strategies conclude with a final epoch in which the full dataset is reintroduced for revision, producing a sharp spike back to 50,000 samples. This U-shaped trajectory reflects the design principle of DBPD: prioritize hard examples early, reduce redundant computation mid-training, and consolidate knowledge through a final full-dataset pass.

Table 10: Accuracy (%) and effective epochs reported as *accuracy/effective epochs* for CIFAR-100 with EfficientNet-B0, trained for 25 and 50 epochs from scratch.

| Model | Baseline | DBPD 0.3 | DBPD 0.7 | SRD 0.95 | SMRD 0.3 | SMRD 0.7 |
|---|---|---|---|---|---|---|
| (50 epochs) EfficientNet-B0 (C100) | 63.65 / 50 | 67.65 / 14.48 | 67.62 / 18.64 | 68.32 / 24.8 | **69.26** / 16.76 | 67.44 / 21.8 |
| (25 epochs) EfficientNet-B0 (C100) | 63.32 / 25 | **67.22 / 10.8** | 66.94 / 14.02 | 17.36 / 17.36 | 67.17 / 11.56 | 66.18 / 14.62 |

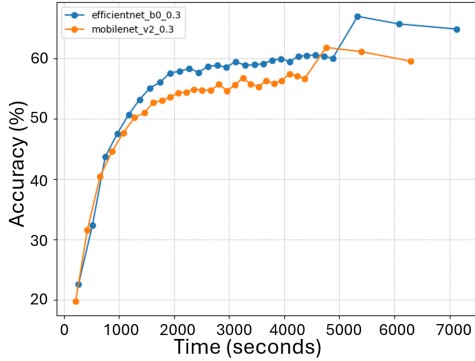

Figure 3: We evaluate EfficientNet and Mo-bileNet by adding more than 1 epoch worth of revision. We see that after an initial boost, there is a drop in performance.

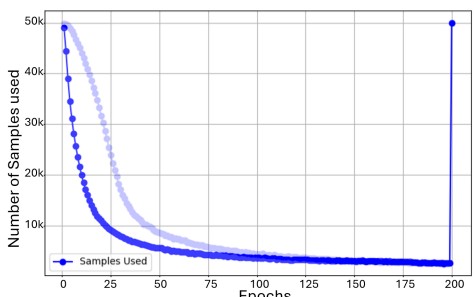

Figure 4: Here, we notice an exponential decay in the number of samples being picked at each epoch for backpropagation. The last epoch includes all samples again. The lighter plot denotes threshold 0.7 while the darker one represents 0.3.

## 6  Limitations

While Progressive Data Dropout demonstrates significant improvements in training efficiency and generalization, it has certain limitations. The method is most effective when training models from scratch; in fine-tuning or transfer learning scenarios, the relative gains are slightly lower. Whilst there are still some gains, pre-training helps to reduce that. Our current evaluation is limited to classification tasks, with preliminary extensions. However, broader applicability to other domains such as segmentation, language modeling, or multi-modal learning remains to be investigated. Additionally, we observe that larger datasets generally require more effective epochs and show lower gains, which may limit some of the training savings. Finally, the method introduces hyperparameters such as thresholds or decay rates, that can be sensitive to the dataset and model. While our default settings perform well across multiple benchmarks, optimal tuning may be necessary in more specialized contexts. Future work would be coming up with mathematical approximations to avoid the use of difficulty based scheduling and adapting the approach to different modalities and tasks.

## 7  Conclusion

We introduced *Progressive Data Dropout*, a simple yet effective approach for improving training efficiency by progressively reducing the amount of data used in each epoch. Through three variants, difficulty-based, scalar-based, and schedule-matched random dropout, we demonstrated that substantial computational savings can be achieved without sacrificing, and in many cases even improving, generalization performance. Notably, the schedule-matched random variant consistently outperformed more structured approaches, highlighting the surprising strength of stochastic data reduction. Our method reduces effective epochs to as little as 12.4% of standard training, with accuracy gains of up to 4.82%. These results suggest that thoughtful data dropout scheduling can serve as a powerful tool for accelerating training in resource-constrained settings, without the need for architectural changes or complex model tuning. We hope this work inspires further exploration of data-centric regularization strategies.

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

## Broader Impact

This work proposes a simple and general-purpose strategy for improving the training efficiency of deep neural networks through selective data dropout. The proposed methods, difficulty-based, random, and schedule-matched dropout, reduce the number of training examples that undergo backpropagation while maintaining or improving final model accuracy. This can lead to significant reductions in computational cost and energy consumption during training.

**Positive societal impact.** By reducing training cost without modifying model architectures or optimizers, our approach lowers the barrier to entry for researchers and practitioners with limited access to high-performance computing resources. This has the potential to democratize access to deep learning research and make machine learning more environmentally sustainable. Furthermore, the method is easy to integrate with existing training pipelines and is applicable to a wide range of model families and datasets, promoting broader adoption of energy-efficient training practices.

**Potential risks.** Our experiments are currently focused on image classification tasks using benchmark datasets. In more sensitive or imbalanced domains (e.g., healthcare or fairness-critical applications), selectively dropping data, even temporarily, could unintentionally bias model learning or overlook rare but important examples. Although our dropout methods reintroduce the full dataset at the end of training, further work is needed to ensure that performance is equitable across subgroups in real-world settings. Additionally, while our method offers efficiency gains, it does not guarantee robustness against adversarial data distributions.

**Environmental impact.** The method is designed to reduce the number of effective epochs required to train a model to convergence. We report gains of up to 10x reduction in backpropagation steps, which translate to real-world energy savings, especially in large-scale training scenarios. This aligns with growing interest in making machine learning more environmentally responsible.

In summary, our work offers a promising and practical step toward efficient and accessible deep learning. We encourage future work to investigate its impact on underrepresented data groups and its integration with fairness-aware or privacy-preserving training pipelines.

## A    Results for 0.7 variants

In the main paper, we report results on the 0.3 variants since that is our best performing variant in most cases along with being more efficient in terms of effective epochs. For completeness, we also report results for the 0.7 threshold in Table 11. Again, we report results as Accuracy/Effective Epochs.

Table 11: Accuracy (%) and effective epochs for threshold 0.7 variants. * refers to the model being pre-trained on ImageNet, else the model is trained from scratch.

| Model | DBPD 0.7 | SMRD 0.7 |
|---|---|---|
| EfficientNet-B0 (C10) | 90.01 / 9.16 | 89.87 / 9.72 |
| EfficientNet-B0 (C100) | 68.80 / 33 | 70.75 / 46.4 |
| *EfficientNet-B0 (C100) | 84.42 / 41 | 84.67 / 51 |
| MobileNet-V2 (C10) | 87.44 / 9.08 | 87.62 / 10.42 |
| MobileNet-V2 (C100) | 63.31 / 40.8 | 65.73 / 55.2 |
| *MobileNet-V2 (C100) | 74.58 / 40.6 | 75.23 / 47.2 |
| ResNet-50 (C10) | 88.20 / 7.12 | 87.01 / 9.8 |
| ResNet-50 (C100) | 60.22 / 21 | 62.62 / 37.6 |
| *ResNet-50 (C100) | 80.51 / 35.6 | 81.45 / 42.8 |
| EfficientFormer-L1 (C10) | 80.66 / 15.52 | 80.81 / 15.62 |
| EfficientFormer-L1 (C100) | 58.08 / 39 | 59.85 / 62.4 |
| *EfficientFormer-L1 (C100) | 86.89 / 59 | 87.78 / 76.8 |
| *ViT-B-MAE (C100) | 86.94 / 71.8 | 87.45 / 82.2 |

## B    Algorithm

We provide a unified training algorithm (Algorithm 1) that captures all three variants of Progressive Data Dropout: Difficulty-Based Progressive Dropout (DBPD), Scalar Random Dropout (SRD), and Schedule-Matched Random Dropout (SMRD). The shared logic between DBPD and SMRD is highlighted in blue, the SRD-specific logic in violet, and SMRD's additional stochasticity in red.

## C    Mathematical Approximation for SMRD

Among the three variants of Progressive Data Dropout we explore, the Schedule-Matched Random Dropout (SMRD) variant achieves the strongest performance across datasets and architectures. SMRD matches the per-epoch data retention schedule of the difficulty-based method (DBPD), but applies dropout stochastically rather than using sample confidence. This yields both generalization benefits

**Algorithm 1:** Unified algorithm for all 3 variants, the common steps between DBPD and SMRD are in blue, specific steps for SRD is in violet and steps for SMRD is in red

**Input:** Model $f_\theta$ with parameters $\theta$, Training data $\mathcal{D}_{\text{train}}$, test data $\mathcal{D}_{\text{test}}$, Number of epochs $E$, revision start epoch $E_{\text{rev}}$, Confidence threshold $\tau : 0 \leq \tau \leq 1$, Loss function $\mathcal{L}$ Decay factor $: 0 < \gamma < 1$

**for** *epoch* $= 1$ **to** $E$ **do**
    **if** *epoch* $\leq E_{rev}$ **then**
        **if** *SRD* **then**
            $r \leftarrow \gamma^{epoch-1}$
        **end**

        **foreach** *batch* $(\mathbf{X}, \mathbf{y}) \in \mathcal{D}_{train}$ **do**
            **if** *SRD* **then**
                $n \leftarrow$ number of samples in $\mathbf{X}$
                $k \leftarrow [r \cdot n]$
                **if** $k = 0$ **then**
                    continue
                **end**
                Randomly sample $k$ indices: $\mathcal{M} \subseteq 1, ..., n$
            **end**
            **if** *DBPD or SMRD* **then**
                $\hat{\mathbf{y}} = f_\theta(\mathbf{X})$
                **if** $\tau = 0$ **then**
                    $\mathcal{M} \leftarrow \{i \mid \arg\max \hat{y}_i \neq y_i\}$
                **else**
                    $p_i = \max \sigma(\hat{\mathbf{y}}_i)$
                    $\mathcal{M} \leftarrow \{i \mid p_i < \tau\}$
                **end**
            **end**
            **if** $\mathcal{M} \neq \emptyset$ **then**
                **if** *SMRD* **then**
                    Randomly sample $|\mathcal{M}|$ indices from $0, ..., |\mathbf{X}| - 1$
                **end**
                $\theta \leftarrow optimizer(\eta, \theta, \mathcal{L}(f_\theta(\mathbf{X}_\mathcal{M}), \mathbf{y}_\mathcal{M}))$
            **end**
        **end**
    **else**
        **foreach** *batch* $(\mathbf{X}, \mathbf{y}) \in \mathcal{D}_{train}$ **do**
            Standard training: $\theta \leftarrow \theta - \eta \nabla_\theta \mathcal{L}(f_\theta(\mathbf{X}), \mathbf{y})$
        **end**
    **end**
**end**

and practical simplicity, but also introduces a dependency on an external schedule derived from a separate DBPD run.

To eliminate this dependency and make SMRD easier to use and evaluate independently, we propose a family of simple mathematical functions that approximate the number of training samples used at each epoch. These approximations allow us to:

- Generate a complete dropout schedule without running DBPD.
- Integrate SMRD seamlessly into standard data loaders.
- Enable reproducible and lightweight evaluation of training efficiency.

We approximate the SMRD schedule using a function $f(x; \alpha)$, where $x$ is the epoch number and $\alpha$ controls the decay rate. To maintain full-dataset training in the final epoch, we define:

Table 12: Accuracy (%) and effective epochs reported as *accuracy/effective epochs* for CIFAR-100 with EfficientNet-B0, 50 epochs from scratch on various scheduler functions.

| Model | Accuracy | Effective Epochs |
|---|---|---|
| Baseline | 63.65 | 50 |
| Inverse Linear Function | 63.51 | **9.4** |
| Logarithmic Decay | 68.62 | 19.12 |
| SMRD 0.3 | **69.26** | 16.76 |
| SMRD 0.7 | 67.44 | 21.8 |
| SRD 0.95 | 68.32 | 24.8 |

$$f(x) = \begin{cases} \hat{f}(x; \alpha), & \text{if } x < T \\ 50{,}000, & \text{if } x = T \end{cases}$$

Here, $T$ is the total number of training epochs (e.g., $T = 200$), and $\hat{f}(x; \alpha)$ is one of the following parameterized decay functions, each normalized to output a maximum of $50{,}000$ (Using CIFAR-100 as our example) samples:

$$\text{(1) Power-Law Decay:} \quad \hat{f}(x; \alpha) = \frac{1}{x^{\alpha}}$$

$$\text{(2) Exponential Decay:} \quad \hat{f}(x; \alpha) = e^{-\alpha x}$$

$$\text{(3) Logarithmic Decay:} \quad \hat{f}(x; \alpha) = \frac{1}{\log(x + \alpha)}$$

$$\text{(4) Inverse Linear Decay:} \quad \hat{f}(x; \alpha) = \frac{1}{x + \alpha}$$

$$\text{(5) Sigmoid-Complement Decay:} \quad \hat{f}(x; \alpha) = 1 - \frac{1}{1 + e^{-\alpha x}}$$

Each function captures a different decay behavior, and the choice of $\alpha$ allows for close approximation of SMRD thresholds (e.g., 0.3 and 0.7) using a single parameter. Lower $\alpha$ values lead to gentler decay (slower dropout), while higher $\alpha$ values correspond to more aggressive early-stage data reduction.

These approximations not only reduce the reliance on a difficulty-based schedule but also make SMRD deployment fully self-contained and differentiable, facilitating future theoretical analysis and integration into automated training pipelines.

## D   Evaluating the approximations on CIFAR-10 and CIFAR-100

We evaluate these approximations on CIFAR-100 and train EfficientNet-B0 from scratch for 50 epochs. Barring the Inverse Linear Function, we notice improvements over the baseline in all cases. However, SMRD 0.3 still gives us our best results. We do note the performance of Logarithmic Decay and consider more detailed ablations on this for future work. The results can be seen in Table 12.

**Do we still have consistent improvements if we are having a longer schedule?** We conducted additional experiments using longer training runs (200 epochs) on CIFAR-10 with EfficientNet-B0, ResNet-50, and MobileNet-v2 architectures. To ensure fairness, we used uniform optimization parameters across all models. Table 13 show that our method achieves consistent improvements under the longer training schedule.

**What happens if we use strong regularization or augmentation method?** Using stronger regularization or augmentation reduces the impact of the current threshold settings in our method. As mentioned in the paper, we report results with a threshold of 0.3 to achieve a balance between speed and accuracy. However, if accuracy is prioritized, higher thresholds can be used, as demonstrated in the table. These results are based on CIFAR-10 training for 200 epochs, using CutMix style augmentation with the parameter $\alpha$ set to 1.

Table 13: Fine-tuning models pretrained on ImageNet (200 epochs).

| Model | Baseline | DBPD (0.3) | SRD (0.99) | SMRD (0.3) |
|---|---|---|---|---|
| EfficientNet-B0 | 95.55 / 200 | 96.11 / 25 | 96.40 / 84 | **96.83 / 31** |
| ResNet-50 | 96.41 / 200 | 96.73 / 27 | 96.83 / 84 | **97.71 / 33** |
| MobileNet-v2 | 93.91 / 200 | 94.41 / 26 | 95.12 / 84 | **95.55 / 33** |

Table 14: Performance under CutMix augmentation.

| Model | Baseline | DBPD(0.3) | DBPD(0.8) | SRD(0.99) | SMRD(0.3) | SMRD(0.8) |
|---|---|---|---|---|---|---|
| MobileNet v2 | 90.41/200 | 90.96/22 | 91.45/32 | **93.29/84** | 91.41/26 | 93.24/42 |

**How does increasing revision epochs or extending total epochs affect?** As shown in Figure 3, adding more revision epochs causes overfitting and can degrade test accuracy. To validate this further, we tested SMRD with 25, 50, and 75 revision epochs (compared to the standard setting of 1). While these settings brought the accuracy closer to the baseline, they did not yield any improvements and risked overfitting as shown in Table 15. Additionally, we increased the overall number of training epochs to bring the overall computation closer to the baseline. As shown in Table 16, this did not really improve our performance much.

**What happens if the dataset has class imbalance or noisy labels?** It is important to emphasize that PDD fundamentally focuses on computational efficiency rather than robustness to class imbalance or noise. Table 17 presents results for Long-Tail Classification on CIFAR-10 with $\rho = 100$, where $\rho$ denotes the ratio between the sample sizes of the most frequent and least frequent classes. The dataset is generated following the same procedure as LDAM-DRW Cao et al. [2019]. We use Focal Loss Lin et al. [2017], consistent with the baseline, to ensure a fair comparison. From the table, all PDD variants show substantial gains over the baseline, with SMRD (0.3) emerging as the best-performing method under class imbalance, achieving the highest accuracy. However, compared to the standard setting, this configuration requires more EE. We evaluated our method on noisy label classification using the Noisy CIFAR-10 dataset, following the setup from Wei et al. [2021], with a noise rate of 17.3% and random noise type, training for 50 epochs. Table 18 shows that all three PDD variants outperform the baseline in both accuracy and efficiency. Notably, SRD (0.95) achieves the highest accuracy, though with slightly higher EE. SMRD and DBPD provide a better trade-off, offering much lower EE while maintaining strong performance, demonstrating their robustness under label noise.

**What is the ideal dataset characteristics for PDD?** PDD is particularly beneficial for mid-sized to large datasets, where the computational savings are most significant. We also evaluated it on smaller fine-grained datasets, such as CUB and Flowers, and observed consistent computational savings without any loss in accuracy as shown in Table 19. These results highlight PDD's suitability across different dataset scales and complexities.

**How do the three variants perform given the same fixed compute budget?** We conducted a preliminary experiment on CIFAR-100 using ResNet-50, aligning the EE across all methods. All models were fine-tuned from ImageNet pre-training. Table 20 shows that even under equal EE, all PDD variants outperform the baseline, with SMRD remaining the best-performing variant.

Table 15: Results of SMRD by increasing revision epochs.

| Method | Revision Epochs | Accuracy (%) |
|---|---|---|
| SMRD (standard) | 1 | 71.01 |
| SMRD | 25 | 68.55 |
| SMRD | 50 | 66.35 |
| SMRD | 75 | 66.27 |
| Baseline | — | 66.32 |

Table 16: Results of extending total training epochs.

| Method | Total Epochs | Accuracy (%) | Effective Epochs (EE) |
|---|---|---|---|
| DBPD | 200 | **67.15** | 25 |
| DBPD | 300 | 67.09 | 29 |
| DBPD | 400 | 66.28 | 33 |
| DBPD | 500 | 66.59 | 36 |
| SMRD | 200 | **71.01** | 35 |
| SMRD | 300 | 70.82 | 44 |
| SMRD | 400 | 70.75 | 49 |
| SMRD | 500 | 70.99 | 51 |
| Baseline | 200 | 66.32 | 200 |
| Baseline | 300 | **66.45** | 200 |
| Baseline | 400 | 66.37 | 200 |
| Baseline | 500 | 66.35 | 200 |

Table 17: Long-tail classification of ResNet-34 on CIFAR-10.

| Model | Accuracy (%) | Effective Epochs (EE) |
|---|---|---|
| Baseline | 46.04 | 200 |
| DBPD (0.3) | 58.09 | 64 |
| DBPD (0.2) | 54.82 | 55 |
| DBPD (0.1) | 24.28 | 65 |
| SRD (0.95) | 54.28 | 84 |
| SMRD (0.3) | **60.14** | **58** |

Table 18: Results of noisy label classification on Noisy CIFAR-10

| Model | Accuracy (%) | Effective Epochs (EE) |
|---|---|---|
| Baseline | 71.91 | 50 |
| DBPD (0.3) | 74.26 | 9 |
| SRD (0.95) | **75.23** | **18** |
| SMRD (0.3) | 73.76 | 9 |

Table 19: Model performance comparison on smaller fine-grained datasets.

| Model | Dataset | SMRD (0.3) | Baseline |
|---|---|---|---|
| EfficientNet-B0 | Airplane | **80.07 / 9** | 78.06 / 100 |
| EfficientNet-B0 | CUB | **71.03 / 9** | 69.83 / 100 |
| MobileNet-v2 | Airplane | **76.92 / 10** | 73.87 / 100 |
| MobileNet-v2 | CUB | **68.94 / 10** | 67.45 / 100 |

**What is the practical usability of SMRD?** To demonstrate the practical usability of SMRD, we derived simple mathematical approximations of the dropout schedule that closely replicate the oracle SMRD performance (Appendix Section C). Among these, the logarithmic decay approximation provides the closest match. Empirical results in Table 21 show that these approximations still clearly outperform the baselines, confirming their practicality. Moreover, SRD and DBPD also outperform the baselines in most scenarios, further validating their effectiveness and applicability.

**What is the wall-clock time using PDD?** While our primary metric, Effective Epochs (EE), enables hardware-independent comparison, we also provide several wall-clock time savings estimates in Table 22 for reference. These experiments were conducted on a single NVIDIA RTX 4060 GPU with 8 GB of RAM.

## E  Number of samples required for SRD

In our setting, we use a constant decay $\gamma$ that samples at the same rate from the dataset iteratively. This could be formalized as :
$$f(e) = \gamma^e \times N \tag{1}$$
where $e$ is the given epoch and $N$ is the number of samples in the dataset. Since we include all the samples as part of our last (revision) epoch $E$, this would be a piecewise function:
$$f(e) = \begin{cases} \gamma^e \times N & \text{if } 1 \leq e \leq E-1 \\ N & \text{if } e = E \end{cases} \tag{2}$$

## F  Number of samples required for SMRD

For our SMRD setting, the number of samples chosen is equal to the number of samples that are above the set threshold for epochs $1 \leq e < E$ where $E$ is the total number of epochs:
$$f(e) = \#\{x \in \mathcal{D} \mid (p_e(x_i)) < \tau\} \, \forall i = 1, ....N \tag{3}$$
We know that as the number of epochs progresses, the number of samples gradually reduce. Assuming the model's prediction probabilities $p_e(x)$ follows a beta distribution : $p_e(x) \sim Beta(\alpha_e, \beta)$. So, at each epoch the number of training samples would be:
$$f(e) = N \times Pr(p_e(x) < \tau) = N \times F_e(\tau; \alpha_e, \beta) \tag{4}$$

Table 20: Performance of aligning Effective Epochs (EE) across methods on CIFAR-100 (ResNet-50)

| Model | DBPD | SRD | SMRD |
|---|---|---|---|
| ResNet-50 | 79.92 / 21 / 0.3 | 79.11 / 20.8 / 0.93 | **80.61 / 20.3 / 0.23** |

Table 21: Practical Usability of SMRD with Logarithmic Decay Approximation.

| Model | Variant | Accuracy | Effective Epochs |
|---|---|---|---|
| ResNet-50 | SMRD (oracle) | 81.54 | 33 |
| ResNet-50 | SMRD (log decay) | 81.11 | 37 |
| EfficientNet-B0 | SMRD (oracle) | 84.45 | 34 |
| EfficientNet-B0 | SMRD (log decay) | 83.92 | 36 |
| MobileNet-v2 | SMRD (oracle) | 74.73 | 36 |
| MobileNet-v2 | SMRD (log decay) | 74.36 | 38 |

Table 22: Wall clock time savings estimates.

| Model | Baseline | DBPD (0.3) | SRD (0.95) | SMRD (0.3) |
|---|---|---|---|---|
| EfficientNet-B0 (CIFAR-100) | 15.74 hr | 7.89 hr | 6.98 hr | 9.95 hr |
| MobileNet-v2 (CIFAR-100) | 12.62 hr | 7.11 hr | 6.67 hr | 9.51 hr |

where $F_e(\tau; \alpha_e, \beta)$ is the cumulative distribution function (CDF) of the Beta distribution.

# G   MAE pre-training

In the main paper, we report results by pre-training MAE using SRD 0.98 and this effectively resulted in a $16 \times$ cheaper pre-training cost whilst showing extremely competitive fine-tuning performance. We also consider a variant of the DBPD by setting a threshold to the MSE loss of each sample. Specifically, we consider two extremes: a threshold of 0.95 which results in most samples being dropped and a threshold of 0.05 which would drop a sample if the MSE is less than 0.05, which helps keep more samples in the pre-training loop. We do a varying amount of revision epochs. For this case, we only perform downstream image classification on ImageNet and **only for 50 epochs**. Results can be seen in Table 23. 'No MAE' refers to using a plain ViT-B backbone from scratch (these typically need fine-tuning for over 300 epochs). 'DBPD 0.95' ends up stopping pre-training as early as 600 epochs with no samples remaining for the threshold. This is expected as an MSE of 0.95 is very high. Setting such a high threshold naturally results in a very low EE of just 0.69 as most samples are dropped. This also results in poor representations. The result with 1 revision epoch is reflecting that. Matching results of training ViT-B from scratch. Using more revision epochs in this case basically means using MAE for said number of epochs. Using a threshold of 0.05 however is much lower and this results in a much higher EE. However, this also gives us a much higher accuracy, even outperforming the full MAE.

# H   Estimation of Effective Epochs (EE) for Comparative Methods

Several prior methods in dataset-level optimization employ data dropout or resampling strategies but do not directly report the number of times the network effectively sees the data during training. To allow a meaningful comparison, we estimate the *Effective Epochs (EE)*—defined as the cumulative number of data sample passes relative to the full dataset size—for both CIFAR-10 and CIFAR-100.

## H.1   DataDropout on CIFAR-10 and CIFAR-100

The DataDropout method reports removing 1,283 "unfavourable" samples after the first epoch. Assuming 30 training iterations for CIFAR-10 and 200 for CIFAR-100, the EE is calculated as follows:

**CIFAR-10**:

$$\text{EE}_{\text{CIFAR-10}} = \frac{1 \cdot 50,000 + (30-1) \cdot (50,000 - 1,283)}{50,000} = \frac{50,000 + 29 \cdot 48,717}{50,000} \approx 29.26$$

Table 23: Accuracy (%) for ImageNet fine-tuning. 'PT EE' corresponds to Pre-training Effective Epochs.

| Model | PT EE | Accuracy |
|---|---|---|
| No MAE | - | 58.61 |
| Full MAE | 800 | 80.26 |
| SRD 0.98 | 49.95 | 77.89 |
| DBPD 0.95 + Revision 1 | 1.69 | 58.05 |
| DBPD 0.95 + Revision 200 | 200.69 | 70.90 |
| DBPD 0.05 + Revision 1 | 743.02 | 80.56 |
| DBPD 0.05 + Revision 50 | 792.02 | **80.65** |

**CIFAR-100**:

$$\text{EE}_{\text{CIFAR-100}} = \frac{1 \cdot 50,000 + (200 - 1) \cdot (50,000 - 1,283)}{50,000} = \frac{50,000 + 199 \cdot 48,717}{50,000} \approx 195.61$$

### H.2 Importance Sampling on CIFAR-10 and CIFAR-100

The Importance Sampling (IS) method dynamically resamples the entire dataset at each iteration. Thus, the Effective Epochs are equal to the total number of training epochs used.

**CIFAR-10**:
$$\text{EE}_{\text{CIFAR-10}} = 30$$

**CIFAR-100**:
$$\text{EE}_{\text{CIFAR-100}} = 200$$

Table 24 summarizes the reported accuracy and estimated Effective Epochs for each method and dataset.

Table 24: Reported estimated Effective Epochs (EE) for DataDropout and Importance Sampling (IS) on CIFAR-10 and CIFAR-100.

| Method | Dataset | EE (Estimated) |
|---|---|---|
| Baseline | CIFAR-10 | 30 |
| DataDropout | CIFAR-10 | 29.2 |
| IS | CIFAR-10 | 30 |
| Ours | CIFAR-10 | **6.98** |
| Baseline | CIFAR-100 | 200 |
| DataDropout | CIFAR-100 | 195.61 |
| IS | CIFAR-100 | 200 |
| Ours | CIFAR-100 | **28.6** |

Directly comparing accuracy is hard as DataDropOut Wang et al. [2018] and IS Katharopoulos and Fleuret [2018] use different backbones, but our estimations show that the effective epochs saved are extremely marginal whilst their reported improvement over the baseline used is marginal compared to us.

## I  Multiple Runs for testing robustness to random results

To verify the robustness of our results, we check the standard deviation over five trials as reported in table 25. From the results, we can notice a minimal standard deviation of $\approx 0.2515$ with a mean accuracy of $88.058\%$. We run EfficientNet-B0 on the CIFAR10 dataset. We notice similar results across different models and datasets showing the robustness of the approach.

Table 25: Accuracy (%) and Effective Epochs (EE) for five trials done using EfficientNet-B0 with SMRD 0.3 variant on CIFAR10 dataset.

| Trial | EE | Accuracy |
|---|---|---|
| Trial 1 | 6.65 | 88.44 |
| Trial 2 | 6.71 | 87.74 |
| Trial 3 | 6.69 | 88.00 |
| Trial 4 | 6.67 | 88.02 |
| Trial 5 | 6.74 | 88.09 |

