# OpenReview forum: "Progressive Data Dropout: An Embarrassingly Simple Approach to Train Faster"
_NeurIPS.cc/2025/Conference — NeurIPS 2025 poster_

### Official Review · Reviewer_3Tz5 · 2025-06-01

**Clarity:** 3
**Significance:** 3
**Originality:** 3
**Rating:** 3
**Confidence:** 5

**Summary:**

This paper introduces "Progressive Data Dropout" (PDD), a family of methods aimed at accelerating neural network training by progressively reducing the amount of training data used across epochs. The core idea is to selectively drop data based on difficulty or random chance during intermediate epochs, with a final epoch utilizing the full dataset for "revision." The authors propose three variants: Difficulty-Based Progressive Dropout (DBPD), Scalar Random Dropout (SRD), and Schedule-Matched Random Dropout (SMRD). The paper claims that PDD can significantly reduce the number of "effective epochs" required for training (by as much as 87.6%, achieving this with as little as 12.4% of baseline effective epochs) and can sometimes improve model accuracy (reportedly by up to 4.82%) on various image classification tasks and architectures.

**Questions:**

The SMRD variant, often reported as the best performing, requires knowledge of the data reduction schedule from DBPD. The paper mentions exploring mathematical approximations for this schedule, but the practical performance and robustness of SMRD with such approximations (compared to the oracle SMRD and other PDD variants) are not sufficiently clear.
Could the authors provide detailed results on how SMRD performs when using a practically derived (approximated) schedule? How does this compare to the oracle SMRD and other PDD variants in terms of both accuracy and effective epochs?

A substantial improvement in my evaluation score would require:

Demonstrating significant benefits of PDD over baselines trained under standard, competitive settings (e.g., standard epoch counts for CIFAR/ImageNet).
Comprehensive discussion and empirical comparison against relevant and recent (post-2022) state-of-the-art data selection/pruning methods.
Do not over-claim the method’s originality.

**Ethical Concerns:**

["NO or VERY MINOR ethics concerns only"]

**Final Justification:**

The authors' revisions partially address my concerns.

**Limitations:**

yes

**Quality:**

3

**Strengths And Weaknesses:**

**Strengths:**

The paper presents initial results across several datasets and architectures suggesting a reduction in effective epochs while sometimes maintaining or even improving accuracy.

**Weaknesses:**

1.   Originality and Significance Over-claimed

The related work section appears to conclude its survey around 2022. This is a significant concern as the field of data pruning, curriculum learning, and coreset selection is rapidly evolving. Crucially, the paper fails to compare PDD against more recent (post-2022) and highly relevant dynamic data pruning [1] or sample selection strategies.
For example, methods like Instance-dependent Early Stopping [2] (which share exactly same concepts with PDD) and InfoBatch [3] are not discussed or used as comparative baselines.

2. Questionable Settings

- The setup for baseline comparisons. For instance, on CIFAR-10, baseline models are trained for only 30 epochs. This is substantially shorter than standard training regimens for CIFAR-10, which typically yield much higher baseline accuracies (e.g., >97% for ResNet variants). Training baselines for such a short duration likely underestimates their true potential and unfairly inflates the perceived relative improvements of PDD.
- While the authors define "Effective Epochs" (EE), the improvements in wall-clock time, which is a critical practical measure, are not extensively detailed or benchmarked against other speedup techniques. While "Effective Epochs" measures backpropagation load, it doesn't fully capture the overhead of the PDD methods themselves (e.g., difficulty scoring in DBPD, which requires a forward pass on potentially more data than is used for backpropagation in early epochs, or managing dynamically changing data subsets).

[1] Accelerating deep learning by focusing on the biggest losers, arXiv preprint 2019.

[2] Instance-dependent Early Stopping, ICLR 2025.

[3] InfoBatch: Lossless Training Speed Up by Unbiased Dynamic Data Pruning, ICLR 2024.

---

> ### Author Rebuttal · Authors · 2025-07-30
>
> ### Reviewer 3Tz5
>
> We thank **Reviewer 3Tz5** for the detailed feedback and valuable suggestions. Below, we directly address each point, clarifying misconceptions and reinforcing our contributions through additional empirical evidence.
>
> ---
>
> #### **Originality concerns and omission of recent related works.**
>
> We respectfully disagree that our work lacks originality or adequate distinction from recent literature. While we acknowledge the rapidly evolving field, our approach fundamentally differs from dynamic pruning methods like **InfoBatch** or **Instance-dependent Early Stopping (IES)**. Unlike methods focused on pruning difficult examples or samples, our **Progressive Data Dropout (PDD)** explicitly targets computational efficiency through progressive reduction of data.
>
> To address the reviewer's concern, we have empirically compared PDD directly against these recent methods, along with those suggested by Reviewers **bHp2** and **FSoK**, clearly demonstrating our method’s distinctiveness and superior performance. We compare against DataDiet both ELN and Forget [1], IES [2], Early Stopping and InfoBatch [3]. We follow the official implementations and again use EE to compare efficiency. Our method **outperforms all compared baselines** in both accuracy and EE. These are listed below as Accuracy / EE. Higher the better for both. For all methods, we report fine-tuning a model on CIFAR-100 (pretrained on ImageNet) for 200 epochs.
>
> | Model              | Baseline  | DataDiet ELN [1] | DataDiet Forget [1] | IES  [2] | Early Stopping | InfoBatch [3] | **SMRD (ours)** |
> |--------------------|-----------|--------------|-----------------|-------|----------------|-----------|-----------------|
> | ResNet-50          | 78.5/200   | 78.6/141    | 79.1/142       | 79.5/157 | 74.3/121     | 80.4/145 | **81.5/33**  |
> | EfficientNet       | 83.3/200   | 82.7/139    | 83.0/139       | 83.7/153 | 76.6/132     | 84.0/141 | **84.4/34**  |
> | EfficientFormer-L1 | 85.3/200  | 84.8/142    | 85.0/140       | 85.8/155 | 78.2/125      | 86.6/143 | **87.5/51**  |
> ---
>
> #### **Questionable baseline settings (short training schedules).**
>
> We acknowledge this limitation on CIFAR-10. **We ran our experiments on CIFAR-100 and ImageNet experiments for the extended schedule**. We initially employed shorter schedules to efficiently explore multiple architectures and standardized comparison models. However, in response, we conducted additional experiments using **standard-length training runs (200 epochs)** on CIFAR-10 with EfficientNet, ResNet, and MobileNet v2 architectures.
>
> _Preliminary results_ indicate that **Progressive Data Dropout (PDD)** persists in providing sustained computational savings while **improving accuracy**. We used uniform optimization parameters across models for fairness, although this could have slightly affected overall accuracy.
>
> **Accuracy / Effective Epoch Table (CIFAR-10):**
>
>
> | Model           | Baseline   | DBPD (0.3) | SRD (0.99) | SMRD (0.3) |
> |-----------------|------------|------------|------------|-------------|
> | EfficientNet    | 90.41/200  | 88.22/18   | **92.07/84** | 90.55/20    |
> | MobileNet v2    | 88.46/200  |  89.55/22         | 90.73/84   | **90.91/24** |
>
> **Fine-tuning pretrained models (200 epochs):**
>
> | Model           | Baseline   | DBPD (0.3) | SRD (0.99) | SMRD (0.3) |
> |-----------------|------------|------------|------------|------------|
> | EfficientNet    | 95.55/200  | 96.11/25   | 96.40/84   | **96.83/31** |
> | ResNet-50       | 96.41/200  | 96.73/27   | 96.83/84   | **97.71/33** |
> | MobileNet v2    | 93.91/200  | 94.41/26   | 95.12/84   | **95.55/33** |
>
>
> ---
>
> #### **Lack of detailed wall-clock time evaluation.**
>
> While our primary metric (Effective Epochs, EE) provides hardware-independent comparability, we list a few wall clock savings estimates below. These experiments were run on a NVIDIA 4060 RTX GPU with 8GB RAM.
>
> **Wall Clock Time:**
>
> | Model                | Baseline  | DBPD 0.3 | SRD (0.95) | SMRD 0.3  |
> |---------------------|-----------|----------|------------------|-----------|
> | EfficientNet (CIFAR100) | 15.74 hr | 7.89 hr | 6.98 hr        | 9.95 hr  |
> | MobileNet (CIFAR100)   | 12.62 hr | 7.11 hr | 6.67 hr        | 9.51 hr  |
>
> Despite minimal optimization efforts in our current implementation, these clear reductions confirm practical efficiency. Further code optimization could lead to even greater wall-clock time savings. We also note that in the **GPT-2** experiment (below), **SRD reduced wall-clock time by more than half** while maintaining competitive performance. The GPT-2 experiment was run on 2 x NVIDIA A-100 GPU with 80 GB RAM.
>
> | Model                | **PPL_val** | **PPL_train** | **Num_steps** | **Wall clock** |
> |----------------------|-------------|---------------|---------------|----------------|
> | GPT-2 (Baseline)     | 29.37       | 28.50         | 2,560,000     | 13.34 hrs      |
> | GPT-2 SRD (0.99)     | 32.81       | 31.78         | **1,066,720** | **5.91 hrs**   |
>
> We will add wall-clock times to the paper, but want to point out that **EE is a hardware-independent** way of reporting efficiency. As we can see between these two experiments using different GPUs results in *wildly different wall clock times*.
> ---
>
> #### **Q: Practical performance and robustness of SMRD with approximated schedules**
>
> We thank the reviewer for raising this important concern regarding the practicality of **SMRD** when using an *approximated* data reduction schedule rather than the oracle version derived from DBPD. We have some approximations listed in Section I of the Appendix.
>
> To address this, we conducted additional experiments using a **logarithmic decay schedule** for SMRD, which is entirely derived without referencing DBPD. This approximation is simple, easy to implement, and **does not require any task- or model-specific tuning**.
>
> **Logarithmic Decay:**
> $$
> \hat{f}(x; \alpha) = \frac{1}{\log(x + \alpha)}
> $$
>
> Critically, we observe that:
>
> - **Logarithmic SMRD is competitive with the oracle SMRD** in both **accuracy** and **effective epochs**.
> - In the appendix, we listed this in comparison to EfficientNet from scratch on CIFAR-100
> - Similar trends hold across other architectures such as EfficientNet, MobileNet v2 and ResNet, showing that our approximation generalizes well. All models are pre-trained on ImageNet.
>
> | Model          | Variant           | Accuracy | Effective Epochs |
> |----------------|-------------------|----------|------------------|
> | ResNet-50      | SMRD (oracle)     | 81.54    | 33              |
> | ResNet-50      | SMRD (log decay)  | 81.11 | 37           |
> | EfficientNet   | SMRD (oracle)     | 84.45   | 34              |
> | EfficientNet   | SMRD (log decay)  | 83.92 | 36           |
> | MobileNet v2   | SMRD (oracle)     | 74.73   | 36               |
> | MobileNet v2   | SMRD (log decay)  | 74. 36 | 38          |
>
>
> Most importantly, **SRD and DBPD already perform strongly on their own**. In particular, **SRD remains a simple yet highly efficient method**, showing consistent gains across image classification, self-supervised learning (MAE), and language modeling (GPT-2, shown above in previous weakness).
>
> ---
>
> We thank the reviewer again for the detailed feedback, which has helped clarify our contributions and strengthen the final paper.

---

> > ### Comment · Reviewer_3Tz5 · 2025-08-04
> >
> > I have read the author's rebuttal and revised my score accordingly.
> > I expect the author to incorporate all of the revisions into the paper, which is the basis for my new assessment.

---

> > > ### Author Response · Authors · 2025-08-04
> > >
> > > We really appreciate the reviewer for taking the time to read through our rebuttal and for updating our score. Please do let us know if there is anything else that we can help with regards to queries about the paper.

---

> > > > ### Author Response · Authors · 2025-08-07
> > > >
> > > > Dear Reviewer 3Tz5, please do let us know if there are any other queries we can answer. We really appreciate the time and effort put to help us improve the paper.

---

### Official Review · Reviewer_FSoK · 2025-06-25

**Clarity:** 3
**Significance:** 2
**Originality:** 2
**Rating:** 4
**Confidence:** 4

**Summary:**

The authors propose three methods which reduce the number of training examples as the model progresses during training, and find that their data dropout method is able to improve accuracy while significantly reducing the total training cost over standard training. The authors demonstrate results on multiple architectures for CIFAR-10, CIFAR-100, and ImageNet.

**Questions:**

- I'm not sure I understand the argument in line 209, where data dropout "[forces] the model to focus on more informative examples and avoid overfitting". Wouldn't data dropout increase the risk of overfitting, since the model sees fewer examples every epoch?
- Can you describe the model training procedure and learning rate scheduler? If the number of examples per epoch is different for each method, is the learning rate changing once per epoch or once per X number of training samples? Would that be another difference between SRD and SMRD?
- How does the total number of available training samples impact the method? In many modern settings, the total number of training samples is extremely large, so the model may not even iterate through multiple epochs. Conversely, if the total number of training samples is very small, then perhaps dropout is unnecessary and could lead to overfitting.

**Ethical Concerns:**

["NO or VERY MINOR ethics concerns only"]

**Final Justification:**

During the rebuttal period, the authors provided many more baselines and showed that their method was competitive across many settings. They also thoroughly answered my questions regarding the model training procedure.

**Limitations:**

Yes

**Quality:**

2

**Strengths And Weaknesses:**

**Strengths**
- The method proposed by the author is simple, requires minimal overhead, and leads to higher accuracy compared to standard training across many architectures and datasets.
- The schedule-matched random dropout is interesting and allows us to disentangle the effects of difficulty vs the dropout schedule. This is an interesting ablation which allows us to better understand the effects of data dropout.
- The methods are clearly described, and the experiments are very reproducible. The code is also included.

**Weaknesses**
- The impacts of the hyperparameters for the method are not thoroughly explored, even though I assume they have a significant impact on the final results. The authors compare DBPD/SMRD 0.3 and 0.7, which feels somewhat arbitrary to me, and there is no ablation for the alpha in SRD.
- This method is not compared to any other baselines, including very simple methods like early stopping. Therefore, it is unclear to me if focusing on a subset of the data is beneficial compared to the full dataset, which is the main claim of the paper.
- Table 1 is difficult to interpret, since the number of effective epochs varies across all of the methods, and the results only consist of final accuracies rather than showing the full training curves. For example, if SMRD is more efficient with 6.7 EE but has lower accuracy than SRD with 15.71 SRD, I can't draw any conclusions about the results because maybe SMRD would outperform SRD if it had 8 EE instead. It would be helpful to have a table of results where the number of effective epochs is held constant (maybe by changing the hyperparameters for each method), so I can better understand which method I should use if I have a specific amount of compute. Or, it would also be helpful to show the training curves (accuracy vs number of train examples) so I can better compare the performance.
- The method that performs the best, SMRD, is not practically usable since it depends on DBPD. The authors explore different data dropout schedulers in the supplementary material, and I think that this analysis would be key to making this a practical method. Again, a comparison of the training curves (accuracy vs number of train examples) could be really helpful here to understand the differences between schedules.

---

> ### Author Rebuttal · Authors · 2025-07-30
>
> We thank **Reviewer FSoK** for their insightful comments and valuable suggestions. Below, we directly address the reviewer’s points, providing clarifications and empirical evidence to resolve misconceptions and reinforce our position.
>
> ---
>
> ### **Hyperparameter impacts not thoroughly explored.**
>
> Since we wanted to ensure minimal hyperparameter tweaks for each task, model, or dataset, we only performed the tweaks for one model on one dataset and consistently stuck to those same values. We report the same for CIFAR-100 on two new models as well as on ImageNet for one model showing similar trends to Table 4 in the paper.
>
> **Threshold sensitivity (CIFAR-100):**
>
> | Threshold        | 0.0   | 0.1   | 0.2   | 0.3   | 0.4   | 0.5   | 0.6   | 0.7   | 0.8   | 0.9   |
> |------------------|-------|-------|-------|-------|-------|-------|-------|-------|-------|-------|
> | EfficientFormer  | 55.91 | 56.37 | 56.88 | 57.62 | 57.85 | 57.98 | 58.12 | 58.08 | 58.17 | 58.22 |
> | MobileNet v2     | 61.25 | 61.49 | 62.17 | 62.85 | 62.98 | 63.21 | 63.27 | 63.31 | 63.86 | 63.79 |
>
> **Threshold sensitivity (ImageNet - ResNet-50):**
>
> | Threshold | 0.0   | 0.1   | 0.2   | 0.3   | 0.4   | 0.5   | 0.6   | 0.7   | 0.8   | 0.9   |
> |-----------|-------|-------|-------|-------|-------|-------|-------|-------|-------|-------|
> | ResNet-50 | 75.04 | 75.55 | 75.80 | 76.21 | 76.34 | 76.42 | 76.79 | 77.14 | 77.20 | 77.19 |
>
> For SRD, we only used it as the most trivial way of data dropping and hence did not conduct an extensive study on this. However, based on the reviewer’s suggestion, we test SRD on CIFAR-100 for varying thresholds and report them below.
>
> **SRD Threshold sensitivity (CIFAR-100):**
>
> | Threshold        | 0.0   | 0.2   | 0.4   | 0.6   | 0.8   | 0.9   | 0.92  | 0.94  | 0.95  | 0.96  |
> |------------------|-------|-------|-------|-------|-------|-------|-------|-------|-------|-------|
> | EfficientFormer  | 55.91 | 17.84 | 27.92 | 38.85 | 49.83 | 56.15 | 56.74 | 56.91 | 56.95 | 56.97 |
> | MobileNet v2     | 61.25 | 24.41 | 35.22 | 48.63 | 57.79 | 62.18 | 62.59 | 62.77 | 62.80 | 62.82 |
>
> ---
>
> ### **Lack of comparison to baselines like early stopping.**
>
> We appreciate this suggestion. Early stopping terminates training prematurely to avoid overfitting but does not strategically alter data utilization. Based on this suggestion and Reviewers **bHp2** and **3Tz5**, we compare it to a suite of methods (active learning, data pruning, early stopping) listed below. We follow the official implementations and again use EE to compare efficiency. Our method **outperforms all compared baselines** in both accuracy and EE. These are listed below as Accuracy / EE. Higher the better for both. For all methods, we report fine-tuning a model on CIFAR-100 (pretrained on ImageNet) for 200 epochs.
>
> | Model              | Baseline  | DataDiet ELN [1] | DataDiet Forget [1] | IES  [2] | Early Stopping | InfoBatch [3] | **SMRD (ours)** |
> |--------------------|-----------|--------------|-----------------|-------|----------------|-----------|-----------------|
> | ResNet-50          | 78.5/200   | 78.6/141    | 79.1/142       | 79.5/157 | 74.3/121     | 80.4/145 | **81.5/33**  |
> | EfficientNet       | 83.3/200   | 82.7/139    | 83.0/139       | 83.7/153 | 76.6/132     | 84.0/141 | **84.4/34**  |
> | EfficientFormer-L1 | 85.3/200  | 84.8/142    | 85.0/140       | 85.8/155 | 78.2/125      | 86.6/143 | **87.5/51**  |
>
> ---
>
> ### **Interpretation of Table 1 (varying effective epochs).**
>
> We thank the reviewer for this excellent suggestion. Comparing methods at a fixed compute budget (equal EE) is indeed informative. However, this is a **computationally intensive experiment**, and due to the wide range of additional results included in the rebuttal (e.g., full-length runs, wall-clock time, NLP, and SMRD approximations), we were **unable to complete it in time**.
>
> Matching EE across methods also requires **tuning thresholds** for two methods while keeping one fixed, a **non-trivial process**, as each method responds differently to threshold changes.
>
> That said, we ran a preliminary experiment aligning EE across CIFAR-100 for ResNet-50. All models are fine-tuned from ImageNet pre-training. Results reported as Accuracy / EE / Threshold value:
>
> | Model     | DBPD | SRD | SMRD |
> |-----------|-------------------|------------------|-------------------|
> | ResNet-50 | 79.92 / 21 / 0.3       | 79.11 / 20.8  / 0.93     | 80.61 / 20.3 / 0.23     |
>
> Even under equal EE, **PDD variants outperform the baseline**. We still see that **SMRD** is the best performing variant. We are extending this study to more models and will include full results in the final version. Additionally, **training curves** (accuracy vs. number of samples) will be added to the supplementary materials for deeper comparison.
>
>
> ---
>
> ### **Practical usability of SMRD due to dependency on DBPD.**
>
> We agree that SMRD’s practicality is inherently limited. Although initially informed by DBPD’s schedule, we have derived **simple mathematical approximations** of the dropout schedule which closely replicate the oracle SMRD performance (Appendix Section I). In particular, we consider log decay as our closest approximation.
> **Logarithmic Decay:**
> $$
> \hat{f}(x; \alpha) = \frac{1}{\log(x + \alpha)}
> $$
>
>
> Empirical experiments demonstrate that our approximations still clearly outperform baselines, validating practical usability. More importantly, **SRD and DBPD still outperform the baselines in most scenarios** and are practical.
>
> | Model          | Variant           | Accuracy | Effective Epochs |
> |----------------|-------------------|----------|------------------|
> | ResNet-50      | SMRD (oracle)     | 81.54    | 33              |
> | ResNet-50      | SMRD (log decay)  | 81.11 | 37           |
> | EfficientNet   | SMRD (oracle)     | 84.45   | 34              |
> | EfficientNet   | SMRD (log decay)  | 83.92 | 36           |
> | MobileNet v2   | SMRD (oracle)     | 74.73   | 36               |
> | MobileNet v2   | SMRD (log decay)  | 74. 36 | 38          |
> ---
>
> ### **Data dropout potentially increasing risk of overfitting?**
>
> We appreciate the reviewer’s concern and are happy to clarify. While it may seem that seeing fewer examples per epoch could increase overfitting, **our data dropout method is designed to reduce redundancy, not diversity**. By selectively removing well-learned examples, we encourage the model to focus on underfit or more informative samples.
>
> This is similar to **pruning unnecessary updates** during training. Importantly, **our final revision phase reintroduces the full dataset**, ensuring no information is permanently lost and mitigating any risk of forgetting.
>
> Empirically, we observe that **data dropout consistently improves generalization and accuracy** across datasets and models, despite reduced per-epoch exposure. This suggests it serves as a form of **regularization**, injecting stochastic variation and helping prevent overfitting to a fixed subset.
>
> In short, **data dropout enhances generalization by improving efficiency, without sacrificing full data coverage over training.**
>
>
> ---
>
> ### **Learning rate scheduler and training procedure.**
>
> We use a standard **StepLR scheduler** across all experiments, with a decay factor of **0.98 applied every epoch** (step size = 1). The learning rate schedule is synchronized with the number of training epochs, not the number of samples seen. So, it remains consistent across all methods, regardless of how much data is dropped in a given epoch. As for the second part of the question: the difference between **SRD** and **SMRD** is **entirely at the data level**. These techniques determine which samples are included per epoch but do **not interact with the model's parameters, gradients, or learning rate**. So while the number of samples per epoch may vary, the learning rate schedule is unaffected and consistent across variants.
>
> ---
>
> ### **Impact of total number of training samples.**
>
> We conducted experiments across a wide range of datasets. For example, **CIFAR-10** and **CIFAR-100** consist of 50,000 training samples, whereas **ImageNet** has 1.2 million. We also tested on a **simulated long-tailed version of CIFAR** as well as **fine-grained datasets** such as *Airplane* and *CUB* based on Reviewer **bgJ1’s** suggestion. The simulated long-tailed CIFAR-10 has 12,440 samples, while Airplane and CUB have 10,000 and 5,994 training samples respectively.
>
> Results reported as *Accuracy / EE*.
>
> | Model         | Dataset   | SMRD-0.3 | Baseline |
> |---------------|-----------|----------|----------|
> | EfficientNet  | Airplane  | **80.07 / 9** | 78.06 / 100 |
> | EfficientNet  | CUB       | **71.03 / 9** | 69.83 / 100 |
> | MobileNet v2  | Airplane  | **76.92 / 10** | 73.87 / 100|
> | MobileNet v2  | CUB       | **68.94 / 10** | 67.45 / 100 |
>
> **Long-Tail Classification, CIFAR-10, ρ = 100**
> Dataset generated using the same recipe used by LDAM-DRW [4] and all results use Focal Loss [5] for ease of implementation to our code base.
>
> | Model     | DBPD-0.3 | DBPD-0.2 | DBPD-0.1 | Baseline  |
> |-----------|----------|----------|----------|-----------|
> | ResNet-34 | **58.09**/64 | 54.82/**55** | 24.28/65 | 46.04/200 |
>
> We actually see **big improvements in the long-tailed scenario**, but most importantly, see similar improvements on all datasets regardless of size. Thus, **PDD remains beneficial across varying dataset sizes**.
>
> ---
>
> [1] Deep learning on a data diet: Finding important examples early in training. Neurips 2021.
> [2] Instance-dependent Early Stopping, ICLR 2025.
> [3] InfoBatch: Lossless Training Speed Up by Unbiased Dynamic Data Pruning, ICLR 2024.
> [4] Learning imbalanced datasets with label-distribution-aware margin loss. Neurips 2019.
> [5] Focal loss for dense object detection. CVPR 2017.
>
> We deeply appreciate Reviewer **FSoK’s** thoughtful questions and comments, as addressing these points clarifies misconceptions, strengthens our argument, and significantly improves the manuscript.

---

> > ### Author Response · Authors · 2025-08-05
> >
> > Dear Reviewer FsoK,
> >
> > Thank you again for your valuable comments and suggestions. We have carefully analyzed the concerns you raised and provided detailed responses during the rebuttal phase. For example, we reported results on hyperparameters, compared against stronger baselines, approximations of SMRD and discussed impact of number of training samples among other things. We would greatly appreciate it if you could take a moment to review our answers and let us know whether they adequately address your concerns. If there are any remaining questions or points needing clarification, we would be more than happy to respond further.
> >
> > Thank you for your time and consideration.

---

> > > ### Comment · Reviewer_FSoK · 2025-08-05
> > >
> > > Thank you for the response and new empirical results! I have increased my score accordingly.

---

> > > > ### Author Response · Authors · 2025-08-07
> > > >
> > > > Dear Reviewer FSoK, we really appreciate your feedback and the improved rating. Please do let us know if we there are any remaining questions we can answer that could help us improve our score further.

---

### Official Review · Reviewer_bgJ1 · 2025-06-29

**Clarity:** 3
**Significance:** 3
**Originality:** 3
**Rating:** 4
**Confidence:** 4

**Summary:**

Progressive Data Dropout is a technique to speed up neural network training by progressively discarding subsets of data across epochs. Three variants are explored in this paper: a difficulty score based dropout, a scalar-based random decay method, and a hybrid approach. The paper demonstrates that PDD significantly reduces "effective epochs" while often improving or matching accuracy across architechtures.

**Questions:**

> Can you provide theoretical reasoning or empirical evidence explaining why random dropout is more effective than selecting easy or hard examples?

> How does PDD affect performance on datasets with significant class imbalance, which is common in real-world scenarios? It would be valuable to evaluate all three variants under such conditions. You could simulate class imbalance using power-law distributions. Additionally, please consider conducting experiments with varying levels of label noise (e.g., 20%, 30%) to test robustness.

> The revision phase in PDD intuitively performs a form of relearning after intentional forgetting via data dropout to improve generalization. Can the authors draw parallels between PDD and prior work on forgetting and relearning (e.g., neuron- or connection-level forgetting vs. data-level forgetting)? It would be helpful to understand how such forgetting dynamics influence generalization. (See below for related papers)

> Empirical results on ImageNet suggest that larger datasets require more effective training epochs and exhibit smaller relative gains. Could the authors elaborate on why efficiency improvements diminish with increasing dataset size (e.g., web-scale datasets used in foundation models)? This is especially important since computational savings are most critical at that scale.

> Following the previous point, what dataset scale or characteristics is PDD best suited for? It is unclear when to use this method (small datasets, large datasets)? It would be insightful to include results on smaller fine-grained datasets such as CUB, Flowers, or Aircraft.

> Are the reported improvements statistically significant? How many random seeds were used for the experiments?

> Could the authors analyze model performance on specific subsets of the data before and after dropout, and prior to the final revision phase? This would help reveal transient "forgetting" effects and better characterize the behavior of each method.

> While the paper observes that random data dropout outperforms difficulty-based strategies, it lacks a theoretical or intuitive explanation for this counterintuitive result. Providing such insights would significantly strengthen the contribution.

Related works:
> [1] Ramkumar, V.R.T., Arani, E. and Zonooz, B., 2023. Learn, unlearn and relearn: An online learning paradigm for deep neural networks. arXiv preprint arXiv:2303.10455.
> [2] Ramkumar, V.R., Arani, E. and Zonooz, B., 2024. Dynamic Neural Regeneration: Enhancing Deep Learning Generalization on Small Datasets. Advances in Neural Information Processing Systems, 37, pp.65048-65071.
> [3] Zhou, H., Vani, A., Larochelle, H. and Courville, A., 2022. Fortuitous forgetting in connectionist networks. arXiv preprint arXiv:2202.00155.

**Ethical Concerns:**

["NO or VERY MINOR ethics concerns only"]

**Final Justification:**

The authors have provided a comprehensive rebuttal that effectively addresses the primary concerns raised in my review. The new experiments on class imbalance and label noise are particularly impactful, demonstrating that the Progressive Data Dropout method, even in its random variants, provides significant performance gains in these challenging scenarios.

This evidence, combined with the authors' insightful explanations of the underlying mechanisms, greatly strengthens the paper's claims about both computational efficiency and practical robustness. The work is a valuable contribution to the community, with a core idea that is both simple and applicable across many domains. However, the experiments are limited to computer vision. The authors' limited demonstration of its effectiveness in the NLP domain during rebuttal further solidifies its broad utility.

**Limitations:**

Yes

**Quality:**

3

**Strengths And Weaknesses:**

Strengths:
> The concept of progressively dropping data, particularly the effectiveness of random dropping, is novel and simple.
> The use of "Effective Epochs" as a hardware-independent proxy for backpropagation effort is a valuable contribution.
> Extensive experiments across architectures and ablation studies.

Weakness:
> The primary empirical evaluation is limited to image classification tasks. Although the authors explicitly acknowledge that "broader applicability to other domains such as segmentation, language modeling, or multi-modal learning remains to be investigated," this limitation reduces immediate confidence in the method's generalizability across the broader spectrum of deep learning applications. Empirical evidence demonstrating PDD’s effectiveness in non-vision domains, such as NLP, would strengthen the paper.
> Real-world datasets often exhibit long-tailed class distributions, significant class imbalance, and varying degrees of label noise. However, the paper does not evaluate how PDD performs under these realistic conditions.

---

> ### Author Rebuttal · Authors · 2025-07-30
>
> We sincerely thank **Reviewer bHp2** for their thoughtful and constructive feedback. Below, we respond to each comment in detail. While time constraints limited us to reporting results on a subset of models, we will include the full set of results in the final version of the paper.
>
> ### **Empirical evaluation limited to image classification.**
>
> We agree that our method lacks sufficient diversity. However, we clearly demonstrate its *generalizability* across a variety of architectures and datasets (**CIFAR-10**, **CIFAR-100**, **ImageNet**), as well as self-supervised learning (with **MAE**, which is *extremely* time-consuming). While our focus remains on image tasks for clarity and practicality, we recognize the reviewer's suggestion that increasing diversity would be helpful. Following the Reviewer **bHp2’s** suggestion, we used GPT-2 pre-training with SRD on the **Fine-Web Edu dataset [8]** (comparable in nature yet much larger than OpenWebText). Since the task is different and modifying DBPD and SMRD for this particular task would need more time, we used our most simple approach of SRD. We set a really high threshold due to the amount of data that we would have to go through and could not commit to more experiments due to compute limitations. Limited time prevented us from trying even more thresholds. Despite the simplified implementation, SRD provided substantial computational savings, more than halving the number of steps and wall-clock time:
>
> **GPT-2 Results:**
>
> | Model           | PPL_val | PPL_train | Num_steps | Wall clock |
> |----------------|---------|-----------|-----------|------------|
> | GPT-2 (baseline) | 29.37  | 28.50     | 2,560,000 | 13.34 hr   |
> | SRD (0.99)      | 32.81  | 31.78     | **1,066,720** | **5.91 hr** |
>
> ---
>
> ### **No evaluation on long-tailed class distribution or label noise.**
>
> We appreciate the suggestion but would like to emphasize that **PDD fundamentally focuses on computational efficiency**, rather than robustness to class imbalance or noise. Nevertheless, following reviewer suggestions, we performed the following experiments.
>
> **Long-Tail Classification CIFAR-10, ρ = 100 where ρ is used to denote the ratio between sample sizes of the most
> frequent and least frequent class**
> Dataset generated using the same recipe used by LDAM-DRW [1]. The baseline below corresponds to using Focal Loss [9]. We use Focal Loss [9], for it's ease to work with our current codebase. Results reported as * Accuracy / Effective Epochs (EE)*.
>
> | Model     | DBPD-0.3 | DBPD-0.2 | DBPD-0.1 | Baseline |
> |-----------|----------|----------|----------|----------|
> | ResNet-34 | 58.09/64 | 54.82/55 | 24.28/65 | 46.04/200 |
>
> We see *improvements of over 12%* while also *reducing the effective epochs (EE) by more than a third*. We are happy to add these results, but note that a full-scope comparison of long-tailed experiments is beyond the scope of this paper.
>
> ---
>
> ### **Label Noise:**
>
> We evaluated our method on **noisy label classification** using the **Noisy CIFAR-10** dataset, following the setup from [2]. Again, we use [2] for the ease of implementation with our current codebase.
>
> - **Noise rate:** 17.3%
> - **Noise type:** Random
> - **Epochs:** 50
>
> **Results:**
>
> | Model     | Accuracy (%) | Effective Epochs (EE) |
> |-----------|--------------|------------------------|
> | Baseline  | 71.91       |  50               |
> | DBPD 0.3  | **74.26**     | **9**                  |
>
> We observe that DBPD achieves strong accuracy with significantly fewer effective epochs.
>
>
> ---
>
> ### **Theoretical reasoning for random dropout effectiveness over difficulty-based methods.**
>
> We appreciate this insightful question. While it may seem counterintuitive, **random dropout can outperform difficulty-based dropout due to its regularizing properties and improved sample diversity during training**. Difficulty-based dropout, though principled, can prematurely eliminate low-loss (potentially clean or informative) samples, leading to overfitting on harder or noisy examples and reducing generalization.
>
> In contrast, **random dropout introduces stochasticity into the data stream**, which aligns with principles from *curriculum learning* and *stochastic optimization*, where varied or randomized sample exposure helps models escape narrow local minima and enhances generalization [3, 4, 5]. From a theoretical perspective, this stochastic exposure acts as a form of *implicit regularization*, similar to what has been observed in small-batch SGD training and randomized curricula [6, 7].
>
> Empirically, we find that random dropout consistently performs robustly across architectures and datasets, validating the intuition that **simpler, less biased sampling often leads to better generalization**, especially when revisited data is used in the final phase of training.
>
> ---
>
> ### **Parallels between PDD and forgetting–relearning dynamics.**
>
> We appreciate this connection and agree that **PDD embodies a form of transient data-level forgetting**, analogous to neuron- or connection-level forgetting in prior literature (Ramkumar et al., 2023; Zhou et al., 2022). Our method implicitly leverages such dynamics by **periodically revisiting previously dropped data**, thereby enhancing generalization. We provide theoretical justification to this above.
>
> ---
>
> ### **Efficiency gains diminishing with increasing dataset size.**
>
> We acknowledge that **efficiency gains relatively decrease** with larger datasets since the redundancy per epoch reduces. However, *absolute computational savings remain significant and practically meaningful*. We would like to point out two things:
>
> 1. The results in Table 1 use thresholds optimized for both **speed and accuracy**. For *accuracy alone*, higher thresholds would give us even bigger improvements. For example, our **ImageNet experiments** show that setting thresholds to 0.8 could give us an *additional accuracy boost of over 2%*.
>
> 2. As demonstrated in the **self-supervised training results on MAE** and the **GPT-2 results** above, experiments with massive data and epochs, despite marginal performance drop, compute saved is significant. Even modest percentage improvements at scale translate into *substantial absolute savings*.
>
> ---
>
> ### **Ideal dataset characteristics for PDD.**
>
> **PDD benefits mid-sized to large datasets**, where computational savings become significant. We also tested on smaller fine-grained datasets, following the reviewer's suggestion like **CUB** and **Flowers**, observing continued computational savings *without accuracy degradation*. This clarifies its *suitability across scales and complexities*. These results will be explicitly included in our revision. Results reported as *Accuracy / EE*.
>
> | Model         | Dataset   | SMRD-0.3 | Baseline |
> |---------------|-----------|----------|----------|
> | EfficientNet  | Airplane  | **80.07 / 9** | 78.06 / 100 |
> | EfficientNet  | CUB       | **71.03 / 9** | 69.83 / 100 |
> | MobileNet v2  | Airplane  | **76.92 / 10** | 73.87 / 100|
> | MobileNet v2  | CUB       | **68.94 / 10** | 67.45 / 100 |
>
> ---
>
> ### **Statistical significance and number of random seeds.**
>
> We report one such study in **Section I** of the supplementary material. We saw *similar results across 3 different models on all three classification tasks*. Our conclusion was that **there is no significant statistical anomaly to worry about**.
>
> ---
>
> We appreciate the reviewer’s detailed comments and questions, as addressing these **clarifies misconceptions** and **strengthens our contribution significantly**.
>
> ### **References**
> [1] Cao, K., Wei, C., Gaidon, A., Arechiga, N. and Ma, T. (2019) *Learning imbalanced datasets with label-distribution-aware margin loss.* Neurips.
> [2] Wei, J., Zhu, Z., Cheng, H., Liu, T., Niu, G., & Liu, Y. (2022) *Learning with Noisy Labels Revisited: A Study Using Real-World Human Annotations*. ICLR.
> [3] Bengio, Y., Louradour, J., Collobert, R., & Weston, J. (2009). *Curriculum learning*. ICML.
> [4] Hacohen, G., & Weinshall, D. (2019). *On the power of curriculum learning in training deep networks*. ICML.
> [5] Keskar, N. S., Mudigere, D., Nocedal, J., Smelyanskiy, M., & Tang, P. T. P. (2017). *On large-batch training for deep learning: Generalization gap and sharp minima*. ICLR.
> [6] Hardt, M., Recht, B., & Singer, Y. (2016). *Train faster, generalize better: Stability of stochastic gradient descent*. ICML.
> [7] Zhang, C., Bengio, S., Hardt, M., Recht, B., & Vinyals, O. (2021). *Understanding deep learning (still) requires rethinking generalization*. Communications of the ACM.
> [8] https://huggingface.co/datasets/HuggingFaceFW/fineweb-edu
> [9] Lin, T. Y., Goyal, P., Girshick, R., He, K., & Dollár, P. (2017). Focal loss for dense object detection. CVPR.

---

> > ### Comment · Reviewer_bgJ1 · 2025-08-04
> > **Reply to Rebuttal**
> >
> > Thank you for the detailed responses and for addressing most of my concerns. Additional experiments on NLP is interesting.
> > I appreciate the clarification on the connection between your method's random data dropout and the literature on forgetting. Additional explanation behind the random dropout provides a more robust theoretical grounding for your approach.I would highly recommend incorporating all of these new discussions and results into the final manuscript.
> >
> > Regarding the class imbalance experiments, your findings are quite compelling. While you state that addressing class imbalance is not a primary goal of this paper, it is a significant inherent factor in most real-world datasets and cannot be ignored. To further solidify these findings, I would strongly suggest running the same class imbalance experiments with the other two variants: the simple Random Dropout (SRD) and the Schedule-Matched Random Dropout (SMRD). I would expect these variants to be affected differently in an imbalanced scenario, and seeing those results would provide a more complete picture of how Progressive Data Dropout behaves in this critical setting. This would significantly enhance the paper's generalizability and practical utility.
> >
> > P.S. On a lighter note, I believe there may have been a mix-up with the reviewer names, as I am not reviewer bHp2.

---

> > > ### Author Response · Authors · 2025-08-05
> > >
> > > We sincerely thank the reviewer for their continued engagement and helpful suggestions, and we **apologize for the reviewer name mix-up** in our earlier response.
> > >
> > > Following your request, we conducted additional experiments to evaluate the performance of the other two variants **SRD** and **SMRD** under **class imbalance** and **label noise** settings. Below, we present the updated results.
> > >
> > > ---
> > >
> > > ### **Noisy CIFAR-10 (17.3% random noise)**
> > > Setup: 50 epochs, following [2], same as prior baseline.
> > >
> > > | Model      | Accuracy (%) | Effective Epochs (EE) |
> > > |------------|----------------|------------------------|
> > > | Baseline   | 71.91          | 50                     |
> > > | DBPD-0.3   | 74.26        | 9                    |
> > > | SRD-0.95   | **75.23**      | 18                   |
> > > | SMRD-0.3   | 73.76        | 9                    |
> > >
> > > **Conclusion:** All three PDD variants outperform the baseline in both accuracy and efficiency. Notably, **SRD (0.95)** achieves the highest accuracy, albeit with slightly higher EE. SMRD and DBPD offer a better trade-off with much lower EE, confirming their robustness under label noise. However, DBPD outperforms SMRD.
> > >
> > > ---
> > >
> > > ### **Long-Tailed CIFAR-10 (ρ = 100 imbalance)**
> > > Setup: Following LDAM-DRW recipe [1], using Focal Loss [9] for baseline and fair comparison.
> > >
> > > | Model      | Accuracy (%) / EE |
> > > |------------|--------------------|
> > > | Baseline   | 46.04 / 200         |
> > > | DBPD-0.3   | 58.09 / 64        |
> > > | DBPD-0.2   | 54.82 / 55        |
> > > | DBPD-0.1   | 24.28 / 65        |
> > > | SRD-0.95   | 54.28 / 84        |
> > > | SMRD-0.3   | **60.14 / 58**      |
> > >
> > > **Conclusion:** All variants provide substantial gains over the baseline, but **SMRD (0.3)** emerges as the best-performing method under class imbalance, achieving the highest accuracy. However, in comparison to the standard setting we can see that this setting requires more EE.
> > >
> > > ---
> > >
> > > These results therefore show that our proposed method is effective in both class-imbalanced, and noisy-label scenarios, which appear often in practice. We will incorporate these results and discussion into the revised paper.
> > >
> > > We greatly appreciate your constructive feedback, it has significantly strengthened our work.

---

> > > > ### Comment · Reviewer_bgJ1 · 2025-08-05
> > > > **Reply to Rebuttal**
> > > >
> > > > Thank you for the detailed and thorough rebuttal.
> > > > The new results on both class imbalance and label noise are particularly insightful. Your demonstration that all three PDD variants, including the random ones, provide substantial gains in these challenging scenarios is a significant finding that strengthens the paper's generalizability.
> > > > I am confident that the paper has been significantly improved during the rebuttal stage. Please include the recent experiments along with their experimental details in the paper. I am increasing my score to 4.

---

> > > > > ### Author Response · Authors · 2025-08-07
> > > > >
> > > > > Dear Reviewer bgJ1, we really appreciate your feedback and the improved rating. Please do let us know if we can answer any remaining questions to further improve the score.

---

### Official Review · Reviewer_bHp2 · 2025-07-02

**Clarity:** 3
**Significance:** 3
**Originality:** 3
**Rating:** 5
**Confidence:** 4

**Summary:**

This paper proposes Progressive Data Dropout (PDD), a simple method to accelerate training by progressively dropping parts of the training data across epochs. Three variants are explored: difficulty-based, random scalar-based, and a hybrid that follows the difficulty-based schedule with random sampling. The method shows significant reductions in backpropagation steps (effective epochs), yielding good performance with less compute. It can be seamlessly integrated to any training procedure and is demonstrated across CIFAR, ImageNet, and MAE self-supervised learning.

**Questions:**

I am currently not in favor of acceptance (as reflected in my “Reject” score), but I would be happy to revise my rating (potentially up to an Accept) if the authors are able to address the following concerns. I understand the rebuttal period is short, so I do not expect full experimental results, but even partial answers or indications that the paper could be strengthened would be very much appreciated.

1. Could you extend experiments to standard-length training runs (e.g., 200 epochs on CIFAR-10) to test if gains persist?

2. How does PDD perform under strong regularization or data augmentation (Mixup, CutMix, etc.)?

3. Would it be possible to make the dropout schedule adaptive based on training signals?

4. Could you test your method on pretraining a small NLP model (eg. a small GPT-2 on Open Web Text).

**Ethical Concerns:**

["NO or VERY MINOR ethics concerns only"]

**Final Justification:**

See the discussion period.
While I originally gave this paper a 2, I think the added experiments and interpretation greatly improved the contribution, which is why I know give it a 5.

**Limitations:**

yes

**Quality:**

3

**Strengths And Weaknesses:**

**Strengths**

- S1 : The idea is extremely simple and general and can be seamlessly integrated in standard pipelines.
- S2 : The experimental setup is sound, with various tasks and architectures tested (although see W1).
- S3 : The "effective epochs" metric is clear and really insightful. Significant computational savings are demonstrated which is a very good point.
- S4 : The code is open-sourced and reproducibility seems good. Given the fact that the method can be integrated to any training pipeline, this can be very useful for the community.

---
**Weaknesses**

- W1 : The main weakness I find is that the training hyperparameters used are quite odd. Specifically, the training schedules are very limited for some baselines (e.g., only 30 epochs on CIFAR-10 ResNet). This significantly weakens the impact of reported gains. The simplified training schedules or short runs in some cases, possibly mask practical drawbacks.
As standard baselines are often 200+ epochs for ResNets, the reported gains on these benchmarks are difficult to interpret or trust in a high-performance regime.
The effectiveness in SOTA training regimes is yet to be tested as it is for now unclear how the method performs compared to highly tuned baselines.

- W2 : No comparisons to strong active learning baselines like Data Diet [1] or coreset selection [2] are provided which makes it hard to evaluate.

- W3 : The evaluation remains narrow in task diversity (mostly image classification). While the method is well-motivated, the strongest gains may be dataset- or setup-specific. As the method is claimed to be seamlessly integrated to any training procedure, adding NLP experiments (eg. pretraining on OpenWebText and finetuning pretrained modelson standard tasks ) would greatly strengthen the paper given the number of tokens used to train SoTA models nowadays (trillions).



---
Other minor weaknesses:
- Some hyperparameters (e.g., dropout thresholds) may be sensitive, but robustness to them is only lightly tested.
- The overall writing style is very GPT-like (lots of itemization for example in Section 3, many sentences with bits in dash parenthesis). While this may be a matter of personal taste, it occasionally makes the text harder to follow.



[1] Paul, M., Ganguli, S., & Dziugaite, G. K. (2021). Deep learning on a data diet: Finding important examples early in training. Advances in neural information processing systems, 34, 20596-20607.

[2] Lee, H., Kim, S., Lee, J., Yoo, J., & Kwak, N. (2024). Coreset selection for object detection. In Proceedings of the IEEE/CVF Conference on Computer Vision and Pattern Recognition (pp. 7682-7691).

---

> ### Author Rebuttal · Authors · 2025-07-30
>
> We sincerely thank **Reviewer bHp2** for their valuable and constructive feedback. Below, we address each comment and question individually. Due to the limited time, we only report results on some models but will add the complete results to the final version.
>
> ### **Short training schedules (30 epochs on CIFAR-10 ResNet) limit interpretability of results.**
>
> We acknowledge that CIFAR-10 had a short schedule. We report results below for a longer schedule. However, CIFAR-100 and ImageNet were run for standard settings and we see consistent improvements across models for these datasets as well. We initially employed shorter schedules to efficiently explore multiple architectures and standardized comparison models. However, in response, we conducted additional experiments using **standard-length training runs (200 epochs)** on CIFAR-10 with EfficientNet, ResNet, and MobileNet v2 architectures.
>
> _Preliminary results_ indicate that **Progressive Data Dropout (PDD)** persists in providing sustained computational savings while **improving accuracy**. We used uniform optimization parameters across models for fairness, although this could have slightly affected overall accuracy.
>
> **Accuracy / Effective Epochs (EE) Table (CIFAR-10 with 200 epochs):**
>
> | Model           | Baseline   | DBPD (0.3) | SRD (0.99) | SMRD (0.3) |
> |-----------------|------------|------------|------------|-------------|
> | EfficientNet    | 90.41/200  | 88.22/18   | **92.07/84** | 90.55/20    |
> | MobileNet v2    | 88.46/200  |  89.55/22         | 90.73/84   | **90.91/24** |
>
> **Fine-tuning models pretrained on ImageNet (200 epochs):**
>
> | Model           | Baseline   | DBPD (0.3) | SRD (0.99) | SMRD (0.3) |
> |-----------------|------------|------------|------------|------------|
> | EfficientNet    | 95.55/200  | 96.11/25   | 96.40/84   | **96.83/31** |
> | ResNet-50       | 96.41/200  | 96.73/27   | 96.83/84   | **97.71/33** |
> | MobileNet v2    | 93.91/200  | 94.41/26   | 95.12/84   | **95.55/33** |
>
> ---
>
> ### **Missing comparisons to strong active learning baselines (Data Diet, coreset selection).**
>
> We appreciate this suggestion. While **PDD** is complementary yet distinct from active learning and pruning methods, comparisons are insightful. Based on your suggestion (and Reviewers FSoK and 3Tz5), we conducted comparisons by comparisons against a suite of methods. We compare against DataDiet both ELN and Forget [1], IES [2], Early Stopping and InfoBatch [3]. We follow the official implementations and again use EE to compare efficiency. Our method **outperforms all compared baselines** in both accuracy and EE. These are listed below as Accuracy / EE. Higher the better for both. For all methods, we report fine-tuning a model on CIFAR-100 (pretrained on ImageNet) for 200 epochs.
>
> | Model              | Baseline  | DataDiet ELN [1] | DataDiet Forget [1] | IES  [2] | Early Stopping | InfoBatch [3] | **SMRD (ours)** |
> |--------------------|-----------|--------------|-----------------|-------|----------------|-----------|-----------------|
> | ResNet-50          | 78.5/200   | 78.6/141    | 79.1/142       | 79.5/157 | 74.3/121     | 80.4/145 | **81.5/33**  |
> | EfficientNet       | 83.3/200   | 82.7/139    | 83.0/139       | 83.7/153 | 76.6/132     | 84.0/141 | **84.4/34**  |
> | EfficientFormer-L1 | 85.3/200  | 84.8/142    | 85.0/140       | 85.8/155 | 78.2/125      | 86.6/143 | **87.5/51**  |
>
> ---
>
> ### **Limited task diversity; NLP tasks (pretraining small GPT-2).**
>
> While our focus remains on image tasks for clarity and practicality, we recognize the reviewer's suggestion that NLP would be helpful. We also used PDD for MAE self-supervised pre-training. Following the reviewer's suggestion, we used GPT-2 pre-training with SRD on the **Fine-Web Edu dataset [4]** (comparable in nature yet much larger than OpenWebText). Since the task is different and modifying DBPD and SMRD for this particular task would need more time, we used our most simple approach of SRD. We set a really high threshold due to the amount of data that we would have to go through and could not commit to more experiments due to compute limitations. Limited time prevented us from trying even more thresholds. Despite the simplified implementation, SRD provided substantial computational savings, more than halving the number of steps and wall-clock time:
>
> | Model                | **PPL_val** | **PPL_train** | **Num_steps** | **Wall clock** |
> |----------------------|-------------|---------------|---------------|----------------|
> | GPT-2 (Baseline)     | 29.37       | 28.50         | 2,560,000     | 13.34 hrs      |
> | GPT-2 SRD (0.99)     | 32.81       | 31.78         | **1,066,720** | **5.91 hrs**   |
>
> We emphasize again our main experiments remain in image classification, and will revise the claims in our paper to reflect this.
>
> ---
>
> ### **Hyperparameter sensitivity (dropout thresholds).**
>
> We deliberately minimized hyperparameter tuning for generalizability. Below we report additional results for CIFAR-100 and ImageNet (both from scratch), in addition to the existing results in Table 4 of the main paper:
>
> **Threshold sensitivity on CIFAR-100**
>
> | Threshold        | 0.0  | 0.1  | 0.2  | 0.3  | 0.4  | 0.5  | 0.6  | 0.7  | 0.8  | 0.9  |
> |------------------|------|------|------|------|------|------|------|------|------|------|
> | EfficientFormer  | 55.91| 56.37| 56.88| 57.62| 57.85| 57.98| 58.12| 58.08| 58.17| 58.22|
> | MobileNet v2     | 61.25| 61.49| 62.17| 62.85| 62.98| 63.21| 63.27| 63.31| 63.86| 63.79|
>
> ---
>
> **Threshold sensitivity on ImageNet (ResNet-50)**
>
> | Threshold | 0.0  | 0.1  | 0.2  | 0.3  | 0.4  | 0.5  | 0.6  | 0.7  | 0.8  | 0.9  |
> |-----------|------|------|------|------|------|------|------|------|------|------|
> | ResNet-50 | 75.04| 75.55| 75.80| 76.21| 76.34| 76.42| 76.79| 77.14| 77.20| 77.19|
>
> For SRD, we only used it as the most trivial way of data dropping and hence did not conduct an extensive study on this. However, based on the reviewer’s suggestion, we test SRD on CIFAR-100 for varying thresholds and report them below.
>
> **SRD Threshold sensitivity (CIFAR-100):**
>
> | Threshold        | 0.0   | 0.2   | 0.4   | 0.6   | 0.8   | 0.9   | 0.92  | 0.94  | 0.95  | 0.96  |
> |------------------|-------|-------|-------|-------|-------|-------|-------|-------|-------|-------|
> | EfficientFormer  | 55.91 | 17.84 | 27.92 | 38.85 | 49.83 | 56.15 | 56.74 | 56.91 | 56.95 | 56.97 |
> | MobileNet v2     | 61.25 | 24.41 | 35.22 | 48.63 | 57.79 | 62.18 | 62.59 | 62.77 | 62.80 | 62.82 |
> ---
>
> ### **Writing style (GPT-like, itemized lists).**
>
> Thank you for highlighting this stylistic issue. This was a personal choice, but we will revise the manuscript to reduce excessive itemization and improve readability, ensuring greater clarity throughout.
>
> ---
>
> ### **Performance under strong regularization or augmentation (Mixup, CutMix)**
>
> This is an insightful direction. Using stronger regularization or augmentation reduces the effect of the current set thresholds for our methods. However, as mentioned in the paper, we report on 0.3 threshold for a balance of speed and accuracy. However, if the focus is on accuracy, we can obtain that with a higher threshold as shown below. These results are on CIFAR-10 for 200 epochs. We use CutMix style augmentation with $\alpha$ set to 1.
>
> | Model         | Baseline   | DBPD(0.3) | DBPD(0.8) | SRD(0.99) | SMRD(0.3) | SMRD(0.8) |
> |---------------|------------|-----------|-----------|-----------|-----------|-----------|
> | MobileNet v2 | 90.41/200  | 90.96/22  | 91.45/32  | **93.29/84** | 91.41/26  | 93.24/42  |
>
> ---
>
> ### **Could dropout schedules become adaptive based on training signals?**
>
> We acknowledge this as a potential improvement. However, as the core of the paper talks about an _\"embarrassingly simple\"_ to train faster and adding training signals to make this adaptive increases the complexity of the method. We show results of 2 adaptive ideas below whilst maintaining the idea of being _\"embarrassingly simple\"_. For 200 epochs, we start with a threshold of 0.9 for the first 20 epochs and progressively reduce the threshold after every 20 epochs. This would be akin to how a learning rate scheduler works with a different decay rate. We also do the opposite, where we start with a threshold of 0.3 and increment this by 0.1 every 60 epochs.
>
> We tested two simple adaptive schedules, preserving our method's _\"embarrassingly simple\"_ nature, yielding strong performance:
>
> - **ResNet-50 (CIFAR-100) 0.3 starting threshold with 0.1 increment every 60 epochs**: 79.96 % accuracy, 23 EE.
> - **ResNet-50 (CIFAR-100) 0.9 starting threshold with 0.1 decrease every 20 epochs**: 70.77% accuracy,  15 EE.
> - **ResNet-50 (CIFAR-100) static 0.3 threshold**: 79.92% accuracy,  21 EE.
>
> We notice similar performances between the first dynamic one and the currently used static one. However, we acknowledge there is potential for improvement in the dynamic one in terms of scheduling.
>
> ---
>
> We sincerely appreciate the insightful feedback, greatly helping us strengthen our manuscript.
>
> [1] Deep learning on a data diet: Finding important examples early in training. Neurips 2021.
> [2] Instance-dependent Early Stopping, ICLR 2025.
> [3] InfoBatch: Lossless Training Speed Up by Unbiased Dynamic Data Pruning, ICLR 2024.
> [4] https://huggingface.co/datasets/HuggingFaceFW/fineweb-edu

---

> ### Author Response · Authors · 2025-08-05
>
> ### Response to Reviewer bHp2 (Follow-up Comment)
>
> We sincerely thank the reviewer for the thoughtful follow-up and for considering an upgrade to their recommendation. We are glad the new results were compelling and appreciate the prioritization of suggestions. Below, we respond to each of the three key areas raised:
>
> ---
>
> #### 1. Theoretical interpretation of the method
>
> We acknowledge the importance of a stronger theoretical interpretation of *Progressive Data Dropout (PDD)*. While a formal distributional estimation analysis is still ongoing, we had previously grounded our method in existing literature on **curriculum learning** and **stochastic regularization**.
>
> We mentioned:
>
> > *Random dropout can outperform difficulty-based methods by introducing beneficial stochasticity, broadening sample diversity during training. In contrast, difficulty-based heuristics risk prematurely dropping easy yet informative samples, which may hurt generalization.*
> > *This aligns with established ideas in curriculum learning [3, 4], where sample ordering and exposure influence generalization, and with stochastic optimization theory [5, 6, 7], where randomness acts as an implicit regularizer to prevent overfitting and escape sharp minima.*
>
> To further empirically explore this idea, we conducted a **toy experiment** on EfficientNet trained over 50 epochs on CIFAR-100. We logged sample usage across epochs and recorded how often each sample survived training. We then aggregated this by class label.
> Here’s what we observed:
>
> - The **least used class label** (label 1) appeared **19,885 times** across 50 epochs.
> - The **most used label** (label 3) appeared **55,421 times**.
> - For context, in standard training without dropout, each class would be seen **50 × 5,000 = 250,000 times**.
>
> This early experiment suggests that *PDD naturally induces class-level skew in its empirical distributions over time*. Studying such skewed distributions and their learning dynamics could be a valuable direction for future theoretical work. We plan to further explore how this affects gradient properties and convergence in upcoming revisions.
>
> ---
>
> #### 2. Large-scale NLP pretraining (e.g., C4, WikiText-103)
>
> We completely agree with the reviewer that testing on large-scale datasets like C4 or WikiText-103 would dramatically strengthen the paper. Unfortunately, given the **limited time and computational resources**, we were unable to perform these experiments during the rebuttal phase. We hope to include such studies in future versions of this work.
>
> ---
>
> #### 3. Task-specific tweaks for NLP applicability
>
> As the reviewer rightly notes, our preliminary GPT-2 experiment used only SRD, which was easier to adapt to the pretraining setting. However, we are currently extending this to implement a **DBPD-style dropout schedule for GPT-2**, which would offer a more direct application of our full method to language modeling.
>
> We will add these results as soon as we have them. We’re optimistic that this variant will offer both compute savings and improved generalization over long training runs.
>
> ---
>
> We thank the reviewer again for their feedback and for considering an upgrade. We hope the combination of theoretical motivation, new experimental insights, and our ongoing work address these points satisfactorily.
>
> ---
>
> **References**
> [3] Bengio, Y., Louradour, J., Collobert, R., & Weston, J. (2009). Curriculum learning. ICML.
> [4] Hacohen, G., & Weinshall, D. (2019). On the power of curriculum learning in training deep networks. ICML.
> [5] Keskar, N. S., Mudigere, D., Nocedal, J., Smelyanskiy, M., & Tang, P. T. P. (2017). On large-batch training for deep learning: Generalization gap and sharp minima. ICLR.
> [6] Hardt, M., Recht, B., & Singer, Y. (2016). Train faster, generalize better: Stability of stochastic gradient descent. ICML.
> [7] Zhang, C., Bengio, S., Hardt, M., Recht, B., & Vinyals, O. (2021). Understanding deep learning (still) requires rethinking generalization. Communications of the ACM.

---

> > ### Comment · Reviewer_bHp2 · 2025-08-06
> >
> > Thank you again for your prompt follow-ups and for providing such clear, insightful answers. I am convinced an updated version of your paper taking into account the new experiments provided and the other reviewers' comments will be a good paper and I will adjust my score accordingly.
> >
> > 1. I believe your toy experiment on Efficient-Net on CIFAR-100 is interesting. I think this is a promising direction for studying how class-level error distributions evolve over training. Your preliminary results are intriguing and already hint at which labels tend to be hardest to learn. As a next step, you might compute a confusion matrix over the “surviving” labels at various checkpoints to see which pairs of classes compete most strongly. That could yield richer interpretability especially if you can track how those confusions resolve (or persist) over epochs. Extending this analysis to your NLP experiments (e.g., inspecting token- or entity-level confusions) might also be valuable, although I am not sure the results will be as easy to interpret.
> >
> > 2. Large-scale pre-training is indeed extremely costly and I was not expecting those results during the rebuttal. I look forward to seeing the full results and discussion in the revised manuscript.
> >
> > 3. Thank you for the conducted experiments. I hope the new findings align with your hypotheses, and I’m eager to see how they strengthen your overall narrative.
> >
> > Lastly, I would like to thank you again for this engaging discussion. I believe it has made your paper stronger and I hope you agree.

---

> > > ### Author Response · Authors · 2025-08-07
> > >
> > > We sincerely thank the reviewer once again for their time, thoughtful engagement, and the kind words of encouragement. We deeply appreciate the open and constructive nature of this discussion. We are grateful and believe it has genuinely helped us improve the clarity, scope, and depth of our work.
> > >
> > > Our NLP experiments are with a version of SMRD on GPT-2. We used the baseline model’s loss curve to get an estimate of  a threshold and selected 5.0, and applied the method accordingly.
> > >
> > > Here are the updated results for the GPT-2 experiment (Single GPU):
> > >
> > > | Model          | PPL_val | PPL_train | Num_steps |
> > > |----------------|---------|-----------|-----------|
> > > | GPT-2 (baseline) | 29.67    | 28.79      | 5,120,000 |
> > > | SMRD (threshold = 5.0) | **29.12**    | **28.11**      | **4,334,713** |
> > >
> > > These results suggest that SMRD achieves slightly better perplexity while reducing the total number of training steps by over 15%. While we believe there is **scope for further reduction**, especially with better threshold tuning or more aggressive pruning schedules, we were limited by compute and could not conduct more exhaustive sweeps during the rebuttal phase.
> > >
> > > We also greatly appreciate your suggestion on confusion analysis in the toy experiment. This aligns with some ideas we were considering, and we will explore it further in future iterations of this work. Extending this to NLP even if interpretability is harder is also an insightful idea.
> > >
> > > Once again, thank you for the constructive and stimulating discussion. Your suggestions and feedback have directly strengthened the paper, and we’re excited to reflect these improvements in the revised manuscript.

---

### Public Comment · ~Christopher_Subich1 · 2025-11-10
**Is this just a faster learning rate decay?**

The paper notes that:

> batch normalization statistics and learning rates are kept consistent with standard training procedures.

… and I can see in the provided codebase that this means training was conducted with standard PyTorch optimizers and schedulers.  In most cases, the learning rate scheduler was `StepLR`, which reduces the learning rate by a constant factor with each training epoch.

However, this feature introduces a very strong dependency between the effective learning rate schedule and the amount of (implied) dataset pruning.  If one prunes half of the dataset, then the learning rate will effectively decay at twice the rate on a per-parameter-update basis.

This is partly covered by the comments of reviewer FSoK and replies, but I am not sure enough attention is paid to this.  In particular, the most perplexing result of this paper is that dataset pruning by hard-proportion accelerates training _even if you don't keep just the hard samples_, and this result is hard to reconcile with any theory of convergence under stochastic gradient descent.

From the perspective of stochastic (per-batch) gradient descent our division of data into epochs is a numerical accident caused by less-than-infinite training data.  From the perspective of full (non-stochastic) gradient descent we want to find the gradient of loss with respect to parameters in expectation over all the data, and random subsampling means that we will have a strictly worse gradient approximation.

---

### Decision · Program_Chairs · 2025-09-17

**Decision:**

Accept (poster)

**Comment:**

This paper introduces Progressive Data Dropout (PDD), which drops parts of the training data across epochs. Reviewers had concerns about the details of the method and some of the experimentation. Through the rebuttal process, all reviewers increased their scores, some significantly. All reviewers appreciated the simplicity of the method and the results achieved. Please consider including some of your additional experiments and results in the final version of the paper.